# Discovery of *N*-(6-Methoxypyridin-3-yl)quinoline-2-amine Derivatives for Imaging Aggregated α-Synuclein in Parkinson’s Disease with Positron Emission Tomography

**DOI:** 10.3390/cells14141108

**Published:** 2025-07-18

**Authors:** Haiyang Zhao, Tianyu Huang, Dhruva D. Dhavale, Jennifer Y. O’Shea, Zsofia Lengyel-Zhand, Dinahlee Saturnino Guarino, Jiwei Gu, Xuyi Yue, Ying-Hwey Nai, Hao Jiang, Marshall G. Lougee, Vinayak V. Pagar, Hee Jong Kim, Benjamin A. Garcia, E. James Petersson, Chester A. Mathis, Paul T. Kotzbauer, Joel S. Perlmutter, Robert H. Mach, Zhude Tu

**Affiliations:** 1Department of Radiology, Washington University School of Medicine, St Louis, MO 63110, USA; zhaohy@sioc.ac.cn (H.Z.); tianyuhuang@wustl.edu (T.H.); jiwei.gu@yale.edu (J.G.); xuyi.yue@nemours.org (X.Y.); naiy@wustl.edu (Y.-H.N.); haojiang@wustl.edu (H.J.); 2Department of Neurology, Washington University School of Medicine, St Louis, MO 63110, USA; dhavaled@wustl.edu (D.D.D.); jenniferoshea@wustl.edu (J.Y.O.); kotzbauerp@wustl.edu (P.T.K.); perlmutterjoel@wustl.edu (J.S.P.); 3Department of Radiology, Perelman School of Medicine, University of Pennsylvania, Philadelphia, PA 19104, USAdinahlee.saturninoguarino@pennmedicine.upenn.edu (D.S.G.); rmach@pennmedicine.upenn.edu (R.H.M.); 4Department of Chemistry, University of Pennsylvania, Philadelphia, PA 19104, USA; mglougee@gmail.com (M.G.L.); vinayak.pagar@gmail.com (V.V.P.); ejpetersson@sas.upenn.edu (E.J.P.); 5Department of Biochemistry and Biophysics, Perelman School of Medicine, University of Pennsylvania, Philadelphia, PA 19104, USA; heejong@heejong.com (H.J.K.); bagarcia@wustl.edu (B.A.G.); 6Department of Radiology, School of Medicine, University of Pittsburgh, Pittsburgh, PA 15213, USA; mathis@pitt.edu

**Keywords:** α-synuclein aggregates, Parkinson’s disease, radiolabeling, PET imaging, tracer development

## Abstract

The fibrillary aggregation of α-synuclein is a hallmark of Parkinson’s disease (PD) and a potential target for diagnostics and therapeutics. Although substantial effort has been devoted to the development of positron emission tomography (PET) probes for detecting α-synuclein aggregates, no clinically suitable tracer has been reported. The design and synthesis of 43 new *N*-(6-methoxypyridin-3-yl)quinolin-2-amine derivatives and an evaluation of their α-synuclein binding affinity is reported here. Compounds **7f**, **7j**, and **8i** exhibited high affinity for α-synuclein and were selected for ^11^C, ^18^F, ^125^I, or ^3^H radiolabeling. A photoaffinity variant, **TZ-CLX**, structurally related to **7j** and **8i**, demonstrated preferential binding to the C-terminal region of α-synuclein fibrils. PET brain imaging studies using [^11^C]**7f**, [^18^F]**7j**, and [^11^C]**8i** in non-human primates indicated that these three α-synuclein PET tracers penetrated the blood–brain barrier. Both [^11^C]**7f** and [^18^F]**7j** showed more favorable brain washout pharmacokinetics than [^11^C]**8i**. In vitro binding assays showed that [^125^I]**8i** is a very potent α-synuclein radioligand, with K_d_ values of 5 nM for both PD brain tissues and LBD-amplified fibrils; it is also selective for PD tissues versus AD or control tissues. These results strongly suggest that the PET probes based on the *N*-(6-methoxypyridin-3-yl)quinoline-2-amine scaffold have potential utility in detecting α-synuclein aggregates in vivo.

## 1. Introduction

Neurodegenerative diseases such as Alzheimer’s disease (AD) and Parkinson’s disease (PD) are associated with the gradual loss of the structure or function of neurons in the brain. More than 40 million people worldwide are affected by these diseases, and the number of cases is expected to rise with increasing life expectancy rates [1]. Unlike other severe diseases such as cancer, where pharmacological treatments have significantly improved survival rates, most neurodegenerative diseases still do not have an effective cure [2]. Although the true cause of many neurodegenerative diseases is unclear, the aggregation of protein monomers into fibrils, via oligomeric intermediates, is a characteristic of many neurodegenerative diseases; the initial process of protein aggregation may occur years prior to the clinical manifestation of neurodegenerative symptoms [3]. Two different insoluble protein aggregates are recognized as pathological hallmarks in AD: (1) the aggregation of amyloid-beta (Aβ) peptides to form extracellular senile plaques (SPs); (2) the aggregation of the tau protein to form the intracellular neurofibrillary tangles (NFTs) [4]. The discovery of [^11^C]PiB was a major breakthrough for the positron emission tomography (PET) quantification of the Aβ protein in the brain [5]. Today, Vizamyl ([^18^F]flutemetamol), Amyvid ([^18^F]flobetapir), and Neuraceq ([^18^F]florbetaben) are FDA-approved to assess the presence or absence of Aβ plaques in the brain of AD patients [6,7,8]. Radiotracers that target NFT tau protein aggregation in the brain, including Tauvid ([^18^F]flortaucipir), Florquinitau, [^18^F]MK-6240, [^18^F]RO948, and [^18^F]PM-PBB3, have also been extensively evaluated in clinical studies with AD patients [9,10,11].

The aggregation of α-synuclein in the brain is recognized as a pathological hallmark of PD [12]. Quantitative PET measures of α-synuclein aggregation could use this biomarker to diagnose early-stage PD, assess PD progression, and guide the development of therapeutics [13]. However, no potent and selective clinically suitable PET tracers for imaging α-synuclein aggregation in PD have yet been reported. Although [^18^F]BF227 was reported to have binding activity for α-synuclein protein, it binds more potently to Aβ plaques, limiting its utility as a PET radiotracer specific for imaging α-synuclein [14]. Investigators have put tremendous effort into the development and optimization of structurally diverse compounds, with the goal of identifying highly potent and selective α-synuclein ligands that could then be used for PET imaging α-synuclein aggregation in the brain. Only a small number of the radioligands have been evaluated in vivo in non-human primates (NHP) [15,16,17,18]. In vitro and in vivo studies of candidate radiotracers showed low α-synuclein binding affinity, poor binding selectivity for aggregated a-synuclein over Aβ or tau, or slow brain kinetics, all of which prevented translation for further clinical investigations [15,16]. Among the physicochemical properties of ligands, lipophilicity plays an important role in the absorption, distribution, metabolism, and elimination of potential radiotracers for neuroimaging [19]. Molecules that target biomarkers for neurological disease are required to have suitable lipophilicity for penetration of the blood–brain barrier (BBB), with optimal calculated LogD (cLogD) values in the range of 1.5–3.0 [20,21,22,23,24]. Although other properties can affect BBB penetration, molecules with moderate lipophilicity often exhibit good brain uptake. Radiotracers that are too lipophilic often have high non-specific binding with slow brain washout pharmacokinetics from non-target sites [16,25,26,27,28,29,30,31,32].

[^18^F]ACI-12589 was recently reported by AC Immune SA as a potential PET radiotracer for imaging α-synuclein protein aggregates in the human brain [33]. The initial results suggest that the retention of [^18^F]ACI-12589 in two specific brain regions, the basal ganglia and cerebellar white matter, reflects the disease process in patients with multiple system atrophy (MSA). In addition, the distribution of [^18^F]ACI-12589 in the basal ganglia and brain stem of the healthy controls was potentially age-dependent. These findings suggest that [^18^F]ACI-12589 tracer binding in the brain is consistent with the expected patterns of MSA α-synuclein pathology based on clinical symptoms. However, further evaluations of disease and target specificity and potential PET signal retention in α-synuclein-positive dementia with Lewy body (DLB) patients versus expected α-synuclein-negative Alzheimer’s disease (AD), progressive supranuclear palsy (PSP), and spinocerebellar ataxia (SCA) patients are still needed.

The discovery of a new class of *N*-(6-methoxypyridin-3-yl)quinoline-2-amine derivatives with high binding potency and selectivity for α*-*synuclein is reported herein. The photoaffinity variant **TZ-CLX**, structurally close to lead compounds **7j** and **8i**, was synthesized for further investigation of the binding sites of these 6-substituted quinolinyl analogues. **TZ-CLX** exhibited significant selectivity for the C-terminal binding sites of recombinant α-synuclein fibrils, with weak binding for N-terminal binding sites. [^11^C]**7f**, [^18^F]**7j**, and [^11^C]**8i** were radiolabeled for NHP PET brain imaging, while [^125^I]**8i** and [^3^H]**8i** were synthesized for further in vitro characterization of ligand binding profiles for aggregated α-synuclein proteins. Each of the PET radiotracers had good uptake in the NHP brain, with appropriately fast washout pharmacokinetics. The in vitro assays showed that the ligands had potent and selective binding to aggregated α-synuclein proteins. These data suggest that the further optimization and evaluation of this new class of radiotracers could help to identify a promising radiotracer for the in vivo imaging of α-synuclein.

## 2. Materials and Methods

### 2.1. Chemistry

All reagents and starting materials used in this manuscript were obtained from commercial sources and used without any further purification. All dry reactions were conducted under a nitrogen atmosphere in an oven-dried glass apparatus using anhydrous solvents. The yields refer to chromatographically homogeneous materials, unless otherwise stated. The reactions were monitored via thin-layer chromatography (TLC) carried out on pre-coated glass plates of the silica gel (0.25 mm) 60 F_254_ from EMD Chemicals Inc. (EMD Chemicals Inc., Gibbstown, NJ, USA) Visualization was accomplished with ultraviolet light (UV 254 nm) or by shaking the plate in a sealed jar containing silica gel and iodine. The flash column chromatography was performed using Silica Flash^®^ P60 40–63 µm (230–400 mesh) from Silicycle (SiliCycle Inc., Quebec City, QC, Canada). The melting points were determined on a MEL-TEMP 3.0 apparatus (Barnstead Thermolyne, Dubuque, IA, USA). The ^1^H NMR and ^13^C NMR spectra were recorded on a Varian 400 MHz (operating at 400 MHz for ^1^H and 100 MHz for ^13^C NMR) spectrometer (Varian, Inc., Palo Alto, CA, USA). Rotamers are denoted by an asterisk (*). Chemical shifts are reported in parts per million (ppm) and coupling constants *J* are given in Hz (Hertz). Chemical shifts are reported relative to TMS (δ = 0.0) as an internal standard. (Abbreviations used in spectra: s = singlet; d = doublet; t = triplet; q = quartet; m = multiplet; br = broad; dd = double of doublets; dt = doublet of triplets; td = triplet of doublets; qd = quartet of doublets.) High-resolution mass spectra (HRMS) [ESI]^+^ were recorded on a Bruker MaXis 4G Q-TOF mass spectrometer with electrospray ionization source (Bruker Daltonics Inc., Billerica, MA, USA).

General Buchwald–Hartwig Amination Procedure A: Cross-coupling of substituted 2-chloro-(iso)quinoline with 5-amino-2-methoxypyridine

In a 20 mL sealed tube, we added 5-amino-2-methoxypyridine (1.0 equivalent), 2-chloro-quinoline (1.0 equivalent), Pd_2_(dba)_3_ (0.05 mol%), xantphos (0.1 mol%), and Cs_2_CO_3_ (2.0 equivalent). The vessel was evacuated and backfilled with nitrogen (three times), then anhydrous 1,4-dioxane (10 mL) was added. The sealed tube was screw-capped and heated to 110 °C. After stirring for 12 h, the reaction mixture was cooled to room temperature and diluted with ethyl acetate. The reaction mixture was filtrated through a pad of celite and washed with ethyl acetate (10 mL × 3). The filtrate was concentrated. The residue was subjected to column chromatography on silica gel to afford the desired product (Sigma-Aldrich, St. Louis, MO, USA).

2-((6-Methoxypyridin-3-yl)amino)quinolin-6-ol (**7e**, **TZ80-34**)

The synthesis procedure contains two steps:

Step 1: Into a 20 mL sealed tube, the following were added: 5-amino-2-methoxypyridine (0.5 mmol, 1.0 equivalent), 2-chloro-6-(methoxymethoxy)quinoline (1.0 equivalent), tris(dibenzylideneacetone)dipalladium(0) (0.025 mol%), xantphos (0.1 mol%), and cesium carbonate (2.0 equivalent). The vessel was evacuated and backfilled with nitrogen three times before adding anhydrous 1,4-dioxane (10 mL). The tube was then tightly capped and heated to 110 °C. After stirring continuously for 12 h, the reaction mixture was cooled to room temperature and diluted with ethyl acetate, and then filtered with a pad of cellite. The filtrate was diluted with ethyl acetate and washed with brine. The combined organic layers were dried over Na_2_SO_4_ and concentrated.

Step 2: The filtrate was concentrated, and then dichloromethane (DCM) (5.0 mL) was used as the solvent. Trifluoroacetic acid (5.0 mL) was added, and the solution was stirred for 10 h. Subsequently, the solution was concentrated and diluted with ethyl acetate. It was then washed with aqueous sodium bicarbonate. The combined organic layers were further washed with brine, dried over sodium sulfate, filtered, and concentrated. The resultant residue underwent purification through silica gel chromatography, yielding a light-green solid product (82.8 mg, 62% yield).

**TZ-CLX** Synthesis and Characterization

6-(2-(3-(But-3-yn-1-yl)-3H-diazirin-3-yl)ethoxy)-*N*-(6-methoxypyridin-3-yl)quinolin-2-amine **(TZ-CLX).**

In a solution of 2-((6-methoxypyridin-3-yl)amino)quinolin-6-ol (25 mg, 0.093 mmol, 1 equivalent) in DMF (1.5 mL), we added potassium carbonate (38.8 mg, 0.280 mmol, 3 equivalent) and 3-(But-3-yn-1-yl)-3-(2-iodoethyl)-3*H*-diazirine (46.4 mg, 0.187 mmol, 2 equivalent) at room temperature (5 min). The reaction temperature increased to 60 °C and stirred for 18 h. The reaction was quenched with water (10 mL) and DCM (20 mL) and then the layers were separated. The aqueous layer was further extracted 2 times with DCM (20 mL). The combined organic layer was washed with saturated brine solution (50 mL), then the organic layer was dried over Na_2_SO_4_ and concentrated under a vacuum. The crude compound was purified via flash column chromatography (gradient of 0–6% MeOH/DCM) to obtain the desired product **TZ-CLX** in 61% yield. TLC (DCM:MeOH, 95:5 *v*/*v*): R_f_ = 0.20; yellow solid (also see Appendix A).

### 2.2. Alpha-Synuclein Binding Affinity Studies

#### 2.2.1. Determination of α-Synuclein Binding Activity Using Radioactive Competitive Assays with [^3^H]BF2846 and [^18^F]**7j** ([^18^F]**TZ61-44**)

Three concentrations of the test compounds (10 nM, 100 nM, and 1 μM) were selected and individually mixed with [^3^H]BF2846 (∼4 nM). Each mixture was then separately added to 50 nM of recombinant α-syn fibrils in a working buffer (50 mM Tris-HCl, 0.01% bovine serum albumin (BSA)). The mixture (150 μL) was gently mixed via agitation, covered, and then incubated at 37 °C for 1.5 h in a non-binding 96-well plate (Corning Inc., Corning, NY, USA; catalog no. 3605). The mixture was filtered through a Unifilter-96 harvesting system (PerkinElmer Inc., Waltham, MA, USA) and then washed three times with 250 μL of ice-cold buffer containing 10 mM Tris-HCl (pH 7.4), 15 mM NaCl, and 20% EtOH. Next, 50 μL of scintillation cocktail (MicroScint-20, PerkinElmer) was added to the collected filtrate and counted on a Microbeta system (PerkinElmer Inc., Waltham, MA, USA). The total binding was measured in the absence of the tested compounds, and non-specific binding was defined by the presence of 100 nM of unlabeled BF2846 in the working buffer.

For the full curve of α-syn competition binding, [^3^H]BF2846 (∼4 nM) was incubated with fixed concentrations of 50 nM α-syn fibrils and ten concentrations of the tested compounds (0.05−1000 nM). All data points were collected in three individual experiments. The equilibrium inhibition constants, K_i_ values, were obtained from EC_50_ values using the equation K_i_ = EC_50_/(1 + [radioligand]/K_d_) via non-linear regression from GraphPad Prism v.9.3.1 (GraphPad Software, San Diego, CA, USA)

In addition, competition assays were performed with LBD-amplified fibrils and [^18^F]**TZ61-44** with slight modifications. The method used to produce LBD-amplified fibrils is described by Dhavale et al. [34], and the source of the tissue was the Banner Sun Health Research Institute Brain and Body Donation Program of Sun City, Arizona. Each test compound concentration (0.001–1 µM) was individually mixed with [^18^F]**TZ61-44** (2 nM), and the resulting mixture was then added to 25 nM of LBD-amplified fibrils in 30 mM of Tris-HCl buffer (pH 7.4) containing 0.1% BSA. The mix was incubated for 1 h with shaking. The samples were then transferred to Multiscreen FB filter plates (Catalog MSFBN6B50, MilliporeSigma) and washed three times with cold (4 °C) buffer. The glass fiber filters containing fibril-bound radioligands were removed and counted immediately in a PerkinElmer (1450-021) Trilux MicroBeta Liquid Scintillation counter using 150 µL of Optiphase Supermix cocktail (Perkin Elmer). All data points were obtained in triplicate. The data were analyzed using Graphpad Prism software (version 10) to the obtain EC_50_ and K_i_ values by fitting the data to a one-site competition model. The synthesis of the tested compounds is described in the Appendix A.

#### 2.2.2. Determination of Binding Activity for AD Tissues Using [^3^H]PiB

Three concentrations of the tested compounds (10 nM, 100 nM, and 1 μM), [^3^H]PiB (∼32 nM), and 0.5 μg/μL of AD tissue homogenate (tissue ID: 05-215) in Dulbecco’s phosphate-buffered saline (DPBS) were added to each well of a non-binding 96-well plate. Non-specific binding was defined by 1 μM of unlabeled PiB in DPBS. The procedures were performed in the same procedure as the α-syn assay using DPBS instead of the working buffer. The synthesis of the tested compounds is described in the Appendix A.

#### 2.2.3. Binding Affinity Measurements of Radiotracers [^11^C]**7f**, [^18^F]**7j**, [^11^C]**8i**, and [^125^I]**8i** and [^3^H]**8i**

The binding affinity determination (K_d_) was performed using homologous competitive binding assays of radiotracers with recombinant α-synuclein fibrils, LBD-amplified fibrils, PD tissues, and AD brain tissue homogenates. The brain tissue from PD and AD cases was obtained from the Banner Sun Health Research Institute Brain and Body Donation Program of Sun City, Arizona [35].

These direct binding assays used a fixed concentration of either recombinant α-synuclein fibrils, LBD-amplified fibrils, PD, AD tissue, or a radioligand ([^11^C]**7f**, [^18^F]**7j,** [^11^C]**8i** and [^125^I]**8i,** [^3^H]**8i**) and varying concentration ranges of the corresponding homologous non-radiolabeled standard reference compound. In brief, various concentrations of corresponding cold compounds were diluted in 30 mM of Tris-HCl at pH 7.4, with 0.1% BSA. The direct binding assay was carried out by mixing (1) recombinant α-synuclein fibrils, LBD-amplified fibrils, PD, or AD tissue; (2) a radioligand ([^11^C]**7f** or [^18^F]**7j**); and (3) various concentrations of corresponding cold compounds, with incubation at 37 °C for 1 h for [^18^F]**7j** and 30 min for [^11^C]**7f**. After incubation, the bound and free radioligands were separated via vacuum filtration through 1.0 µm glass fiber filters in 96-well filter plates (MilliporeSigma, Burlington, MA, USA), followed by three 200 µL washes with ice-cold assay buffer. The filters containing the bound ligand were mixed with 150 µL of Optiphase Supermix scintillation cocktail (PerkinElmer) and counted immediately. All data points were obtained in triplicate. The dissociation constant (K_d_) and the maximal number of binding site (B_max_) values were determined by fitting the data to the following equation: bound = (B_max_ × [radioligand])/([radioligand] + [unlabeled compound] + K_d_) + bottom. This was achieving via non-linear regression using GraphPad Prism software (version 4.0), where (bottom) is the non-specific binding and the [radioligand], [unlabeled compound], and K_d_ are expressed in nM. The synthesis of the tested compounds is described in the Appendix A.

### 2.3. Radiochemistry

Briefly, the [^11^C]CH_3_I was produced on site from [^11^C]CO_2_ using a GE PETtrace MeI Microlab. Up to 1.4 Ci of [^11^C] carbon dioxide was produced from the JSW BC-16/8 cyclotron by irradiating a gas target of 0.5% O_2_ in N_2_ for 15–30 min with a 40 μA beam of 16 MeV protons in the Barnard Cyclotron Facility of Washington University School of Medicine. After the [^11^C]CO_2_ was converted to [^11^C]CH_4_ using a nickel catalyst (Shimalite-Ni (reduced), Shimadzu, Kyoto, Japan P.N.221-27719) in the presence of hydrogen gas at 360 °C, the [^11^C]CH_4_ was further converted to [^11^C]CH_3_I via a reaction with iodine in the gas phase at 690 °C. Approximately 12 min after the end-of-bombardment (EOB), several hundred millicuries of [^11^C]CH_3_I were delivered in the gas phase to the hot cell where the radiosynthesis was accomplished.

#### 2.3.1. Radiosynthesis of [^11^C]**7f**

Approximately 1.0 mg of precursor **7e** was placed in a V-shaped reaction vessel with an aqueous NaOH solution (5 M) in MeCN (300 µL). The [^11^C]CH_3_I was bubbled into the reaction vessel and the reaction mixture was heated to 80 °C for 5 min. After quenching with 1.8 mL of HPLC mobile phase (48% acetonitrile in 0.1 M ammonium formate buffer, pH 4.5). The reaction mixture was loaded onto a C18 column (Agilent Zorbax SB-C18, 5 µm, 250 × 9.6 mm, Santa Clara, CA, USA), then eluted from the column using the above-mentioned HPLC mobile phase at a flow rate of 4.0 mL/min. A 100 mL glass vial pre-filled with 60 mL of sterile water was used to collect the radioactive product from 12 to 13 min and then passed through a C18 Sep-Pak Plus cartridge with nitrogen gas assistance. The trapped product was eluted using 0.6 mL of ethanol and 5.4 mL of saline to formulate the injection dose. The product was authenticated using an analytical HPLC system (Agilent SB-C18 analytic column, 250 mm × 4.6 mm; mobile phase of 75% acetonitrile in 0.1 M ammonium formate buffer, pH 4.5; flow rate of 1.0 mL/min; UV wavelength of 254 nm; t_R_ = 3.9 min) by co-injecting with the standard reference compound **7f**. The radiochemical yield was about 45%, the radiochemical purity was >99%, and the specific activity was >74 GBq/µmol (decay corrected to EOB).

#### 2.3.2. Radiosynthesis of [^11^C]**8i**

Approximately 1.0 mg of precursor (**10**) was placed in a V-shaped reaction vessel with KOH solid (1 mg) in DMF (300 µL). Then, the mixture was shaken for 1 min using a Vertx mixer. Next, [^11^C]CH_3_OTf was bubbled into the reaction vessel and the reaction mixture was heated to 90 °C for 5 min. After quenching with 1.8 mL of HPLC mobile phase (65% acetonitrile in 0.1 M ammonium formate, buffer, pH 4.5), the reaction mixture was loaded onto a C18 column (Agilent Zorbax SB-C18, 5 µm, 250 × 9.6 mm), then eluted from the column using the above-mentioned HPLC mobile phase at a flow rate of 4.0 mL/min. A 100 mL glass vial pre-filled with 60 mL sterile water was used to collect the radioactive product from 18 to 19 min and then passed through a C18 Sep-Pak Plus cartridge with nitrogen gas assistance. The trapped product was eluted using 0.6 mL of ethanol and 5.4 mL of saline to formulate the injection dose. The product was authenticated using an analytical HPLC system (Agilent SB-C18 analytic column, 250 mm × 4.6 mm; mobile phase of 75% acetonitrile in 0.1 M ammonium formate buffer, pH 4.5; flow rate of 1.0 mL/min; UV wavelength of 254 nm; t_R_ = 4.2 min) by co-injection with the standard reference compound **8i**. The radiochemical yield was about 16%, the radiochemical purity was > 99%, and the specific activity was > 74 GBq/µmol (decay corrected to EOB).

#### 2.3.3. Radiosynthesis of [^18^F]**7j**

A solution of the precursor **7e** (2 mg) and Cs_2_CO_3_ (2.0 mg) in DMSO was added to a reaction vessel containing [^18^F]**12**. The vessel was capped and heated at 100 °C for 15 min. Subsequently, the reaction mixture was diluted with 2.7 mL of HPLC mobile phase (38% acetonitrile in 0.1 M ammonium formate buffer, pH 4.5) and loaded onto a C18 column (Agilent SB-C18, 250 mm × 10 mm), and eluted from the column using the above-mentioned HPLC mobile phase at a flow rate of 4.0 mL/min. A 100 mL glass vial that contained 50 mL of sterile water was used to collect the radioactive product from 25 to 26 min the diluted product was then passed through a Sep-PakPlus C18 cartridge with nitrogen gas assistance. The trapped product was eluted using 0.6 mL of ethanol and diluted in 5.4 mL of saline to formulate the injection dose. The product was authenticated using an analytical HPLC system (Agilent SB-C18 analytic column, 250 mm × 4.6 mm; mobile phase of 70% acetonitrile in 0.1 M ammonium formate buffer, pH 4.5; flow rate of 1.0 mL/min; UV wavelength of 254 nm; t_R_ = 4.8 min) via co-injection with the standard reference compound **7j**. The radiochemical yield was about 35%, the radiochemical purity was > 98%, and the specific activity was > 55 GBq/µmol (decay corrected to EOB).

#### 2.3.4. Radiosynthesis of [^125^I]**8i** and [^3^H]**8i**

In a 2 mL low-retention centrifuge tube, we added 50 µL of tributylstannane precursor **7×** (4 mg/mL) and 50 µL of freshly prepared 5% sodium acetate in glacial acetic acid. Next, 10 µL of Na[^125^I] was added to the vial. After the addition of 50 µL of freshly prepared hydrogen peroxide/acetic acid, the mixture was stirred at room temperature for 15 min and vortexed 3 times. The reaction mixture was then loaded onto a semi-preparative HPLC system for purification. The retention time for the target compound was 17 min. The desired HPLC fraction was diluted with 50 mL of sterile water and an aliquot removed for radio-TLC, before concentration on a Sep-Pak C18 Plus cartridge, the trapped product was washed with 20 mL of water, then eluted in ethanol. Before the Sep-Pak was eluted with ethanol, purity was confirmed to be >95% via radio-TLC (hexane/ethyl acetate: 2:1).

The radioligand [^3^H]**8i** was custom synthesized by Novandi Chemistry AB Forskargatan 20J, Södertälje, SE-15136, Sweden, via a subcontract.

### 2.4. PET Brain Imaging Studies in Non-Human Primates

All animal experiments were conducted under a research protocol approved by the Washington University Institutional Animal Care and Use Committee (IACUC). The NHP study was conducted in the NHP microPET facility at the Washington University School of Medicine in St. Louis. Three adults male cynomolgus macaques (9.73 ± 0.71 kg) were used in this study. Each animal was fasted for 12 h prior to the PET scans. Anesthesia was induced with an intramuscular injection of ketamine (10–20 mg kg^−1^) and glycopyrrolate (0.013–0.017 mg kg^−1^) to reduce salivary gland and respiration secretions. The animal was intubated, and anesthesia was maintained at 40–50% N_2_O and 1.4–2.0% isoflurane/oxygen throughout the procedure. A percutaneous venous catheter was used for the radiotracer injection. During the PET scanning session, the head was positioned supine with the brain in the center of the field of view. The PET scans were performed with a microPET Focus 220 scanner (Concorde/CTI/Siemens Microsystems, Knoxville, TN, USA). A 10 min transmission scan was performed to check the positioning; once confirmed, a 45 min transmission scan was obtained for attenuation correction. Subsequently, a 2 h dynamic emission scan was acquired after the administration of 365.44 ± 26.69 MBq of [^11^C]**7f**, [^11^C]**8i**, or [^18^F]**7j** via the venous catheter. The PET data were collected from 0 to 120 min with the following timeframes: 3 × 1 min, 4 × 2 min, 3 × 3 min, and 20 × 5 min. The PET images were reconstructed using a filtered back projection method into a volume size of 128 × 128 × 95 and a voxel size of 1.85 × 1.85 × 0.796 mm^3^, with all corrections (scatter, decay, random, normalization, etc.) applied. The dynamic PET images were motion-corrected and then co-registered to previous acquired structural MPRAGE MR images. The INIA19 primate brain and atlas (https://www.nitrc.org/projects/inia19/, accessed on 2 August 2023) were then co-registered to the MR images, so as to obtain the regional time−activity curves (TACs) in the regions of interests (ROIs). The radioactivity uptake was normalized using the body weight and the injected dose to obtain the standardized uptake values (SUVs). A voxel-wise reference Logan plot was performed using the cerebellum as the reference region to obtain the distribution volume ratio (DVR) [24], with a linear time range of 30 min to 120 min:(1)$$\frac{\int_{0}^{t}\;C_{t}\left(\tau \right)d\tau }{C_{t}\left(t\right)}=DVR\frac{\int_{0}^{t}\;C_{r}\left(\tau \right)d\tau }{C_{t}\left(t\right)}+\beta$$

Here, C_r_(t) is the regional time activity curve (TAC) of the reference tissue, C_t_(t) is the pixelwise time activity curves of the target tissue tracer concentration measured via PET, and DVR is the distribution volume ratio of the target to the reference tissue.

The image processing was performed using MIAKAT in MATLAB (R2023a, The MathWorks, Inc., Natick, MA, USA) [25].

## 3. Results and Discussion

### 3.1. Design

To develop a new radiotracer for imaging α-synuclein**,** our structure–activity relationship analysis focused first on the exploration of new analogues based on our previous work [15]. We previously reported that [^11^C]**13** and [^18^F]**28** (Figure 1) had a moderate binding potency and lower selectivity for α-synuclein versus Aβ plaques and tau fibrils. In pursuit of enhancing the selectivity for α-synuclein, we implemented modifications by replacing the double bond or oxadiazole bridge with an amino group. This replacement yielded quinolinyl pyridinyl amines, establishing a flexible linkage between the quinolinyl and pyridinyl segments. As described in Figure 2, a series of new *N*-(6-methoxypyridin-3-yl)quinoline-2-amine derivatives were synthesized, and their binding potency and selectivity rates for α-synuclein were determined.

### 3.2. Chemistry

To develop amines in which the *N*-atom serves as a connecting link between diverse quinolinyl moieties and pyridinyl moieties, several innovative strategies were employed. The initial focus was centered on the alteration of the position of the *N*-atom within the quinolinyl ring. This was achieved by coupling various quinolinyl halides **1a**–**g** with 6-methoxypyridin-3-amine **2a** via the Buchwald–Hartwig amination reaction, and seven new compounds **3a**–**g** were synthesized, as shown in the upper section of Figure 1. Furthermore, fluorine (F) was introduced into different positions of the quinoline ring to synthesize amines **5a**–**f**, as also shown in Figure 1, by starting with fluoride-containing heteroaryl halides **4a**–**f**, reacted with **2a**. Through this initial exploration, thirteen new compounds based on the *N*-(methoxyphenyl)quinolinyl-amine pharmacophore structure were synthesized.

The next strategy focused on introducing different functional groups into the 6-position of the quinolinyl ring. Different functional groups such as -NO_2_, -N(CH_3_)_2_, -Br, -OCH_3_, -OCF_2_H, -OCF_3_, -OCH_2_CH_2_F, and -OCH_2_CF_3_ substituted at the 6-position of the quinolinyl-2-halides were reacted with 6-methoxypyridin-3-amine, and seven new compounds **7a, 7b, 7d, 7f, 7g, 7h,** and **7j** were obtained. Compound **7c** was prepared from compound **7a** via the reduction reaction shown in Figure 2. In order to prepare compound **7e**, compound 6-(methoxymethoxy)-N-(6-methoxypyridin-3-yl)quinolin-2-amine (**7x**) was first made via Buchwald–Hartwig amination, followed by the deprotection of the –MOM group using trifluoroacetic acid, and **7e** was afforded as shown in Figure 2. The hydroxyl compound **7e** was a key intermediate for preparing several other molecules. Compound **7e** reacting with fluorosulfuryl imidazolium triflate salt afforded compound **7i**. O-alkylating the phenol group in **7e** using 2-bromo-1,1,1-trifluoroethane afforded compound **7u**. The PEGylating modification strategy is used in drug design to decrease the *c*LogD value, as lower lipophilicity may improve the BBB permeability of a molecule and reduce the non-specific binding of a molecule. In the current study, a PEGylating strategy was investigated for the design and structural modification of α-synuclein ligands. Through the O-alkylation of the phenol group in **7e** using a 1-3 PEGylated unit containing a terminal fluoroethoxy group, compounds **7k**, **7l**, and **7m** were obtained (Figure 2).

In addition, by keeping the methoxy moiety at the 6-position of the quinolinyl moiety, **7n, 7o**, and **7q** were obtained through the Buchwald–Hartwig cross-coupling reaction between 6-methoxyquinolin-2-chloride and pyridin-3-amine, having -F, -CF_3,_ or -OCF_3_ at the 6-position of the pridinyl-3-amine. Meanwhile, to keep the 2-fluoroethoxy group at the 6-position of the quinolinyl moiety, **7p, 7s**, and **7t** were obtained using 6-(2-fluoroethoxy)quinolin-2-chloride to couple with pyridinyl-3-amine, having -CF_3,_ -CH_3_, or -H at the 6-position of the pridinyl-3-amine. The amine **7r** was made using 6-methoxyquinolin-2-amine reacted with 5-bromo-2-(methylthio)pyridine. As shown in Figure 2, a total of twenty 6-substituted quinolinyl-*N*-(6-substitution)pyridine-3-yl-2-mines were synthesized.

To further explore the different substitution groups at the 6-position of the quinolinyl moiety impact on the binding activity for α-synuclein protein, new analogues were synthesized. Briefly, starting with compound **7e** as the key intermediate, via the O-alkylation of the phenol with different substituted benzyl halides or cyclohexyl *p*-toluenesulfonate, nine new compounds **8a–i** were obtained using the nucleophilic reaction conditions depicted in Figure 3. In addition, compound **8j**, which contains two pyridyl groups and one quinolinyl fragment, was synthesized using two equivalents of 6-methoxypyridin-3-amine (compound **2a**), reacted with one equivalent of 6-bromo-2-chloroquinoline 6; as shown in Figure 3, one equivalent of 6-methoxypyridin-3-amine replaced the bromide, and the other equivalent replaced the chloride in **6**. A total of ten new amines that contained either a saturated cyclic ring or unsaturated heteroaromatic ring at the 6-position of the quinolinyl moiety were made for this series of analogues, as shown in Figure 3.

To summarize our efforts on the design and synthesis of new α-synuclein compounds, first the position of the *N*-atom in the quinolinyl ring was changed to identify analogues with *N*-(pyridinyl-3-yl)quinolinyl-2-amines as the pharmacophore structure. Different functional groups were then introduced at different positions of the quinolinyl-2-amino moiety to confirm that the 6-position of the quinolinyl-2-amino moiety is worth exploration. The substitution of different functional groups such as uncyclized and cyclized alkyl groups, aromatic rings, or hetero-aromatic rings at the 6-position of the quinolinyl-2-amino moiety was the next modification strategy. A PEGylated strategy, designed to improve the lipophilicity of potential ^18^F radioligands, was followed to synthesize compounds in which different PEGylated units were introduced between the phenolic group and the 2-fluoroethoxyl group. In addition, analogues using different substituted groups in the ortho-position of the *N*-pyridyin-3-yl ring were also synthesized. In total, we synthesized 43 new amine analogues composed of different substituted quinolinyl and pyridinyl amines; these ligands were then screened to determine their binding potency and selectivity toward α-synuclein.

### 3.3. Bioactivity Determination

#### 3.3.1. Determination of α-Synuclein Activity

The binding potencies for α-synuclein were first determined using recombinant α-synuclein fibrils through a radioactive competitive assay using [^3^H]BF2846, a potent but non-selective α-synuclein radioligand. Additional competitive binding assays were carried out with [^18^F]**7j** as the radioligand and α-synuclein-amplified LBD fibrils as the binding substrate. An initial assessment was carried out for seven new compounds, **3a–g**, which contain a *N*-(2-meothxy pridinyl)-yl ring and a simple quinolinyl ring in which the position of the *N*-atom varied but no other substituted groups were introduced; the in vitro binding of **3a–g** to α-synuclein fibrils is presented in Table 1. The assay results show that the position of the *N*-atom in the quinolinyl ring affected the binding affinity for α-synuclein fibrils. Compound **3a**, which has the *N*-atom at the 1-position of the quinolinyl ring, is the most potent α-synuclein compound, with a K_i_ value of 100 nM, as compared with compounds **3b–g**. Therefore, we next focused on introducing different substituted groups into the 1-*N*-quinolinyl ring. As shown in Table 2, an F-atom was introduced at different positions of the 1-*N*-quinolinyl ring to make six compounds, **5a–f.** Compounds **5a–f** had modest potency, with K_i_ values in the range of 27 to 65 nM for α-synuclein fibrils. Out of these six compounds, **5d**, with an F-atom at the 6-position of the quinolinyl ring, is the most potent compound, with a K_i_ value of 27 nM, suggesting that exploring the 6-position of the quinolinyl ring may lead to discovering new compounds that are favorable to α-synuclein. Therefore, both electronic-withdrawing and -donating groups such as -NO_2_, -NH_2_, -Br, -OH, and -OCH_3_ were introduced to replace the proton (H-) at the 6-position to make compounds **7a–c**, which had modest potency, with K_i_ values of 21 to 82 nM, as shown in Table 2. Compound **7f,** with a methoxy (-OCH_3_) at the 6-position of the quinolinyl ring, was the most potent α-synuclein compound, with a K_i_ value of 4.6 nM. Furthermore, for compounds **7g** and **7h** with -OCF_2_H or -OCF_3_, the bio isosteres of-OCH_3_ yielded K_i_ values of 23 nM for **7g** and 100 nM for **7h**. Compared to **7g**, for compounds **7h** to **7d**, the electron-withdrawing group reduced the α-synuclein binding potency; a much stronger electronic-withdrawing fluorosulfuryl group further diminished the α-synuclein activity, with a K_i_ value of 320 nM for compound **7i**. Compound **7j** with the -OCH_2_CH_2_F group displayed high potency, with a K_i_ value of 6.4 nM.

Building on the structure–activity relationship analysis of the *N*-(quinolinyl)-pyridinyl-2-amines reported in Table 1 and Table 2, new derivatives were synthesized, focusing on optimizing the substitution at the 6-position of the *N*-(6-methoxypyridin-3-yl)quinolinyl-2-amine. Compounds **7k**, **7l**, and **7m** containing 1-3 PEGylating units had K_i_ values of 50.0, 49.0, and 42.0 nM, respectively, for their α-synuclein binding potency. The cLogP values were 4.25, 4.08, and 3.90 for **7k, 7l,** and **7m,** respectively, indicating that PEGylating modification increases the hydrophilicity. The methoxy group in the pyridinyl ring of **7d** and **7j** was replaced by -F, -CF_3_, or -OCF_3_ groups to make compounds **7n–q**; except for compound **7q**, these replacements diminished the α-synuclein affinity, as compounds **7n, 7o, 7p** had only moderate potency, with K_i_ values of 81.0–400 nM, as shown in Table 3. Meanwhile, compounds **7r** and **7s** were synthesized by replacing the methoxy group in **7d** using -SCH_3_ and -CH_3_; both had lower α-synuclein affinities, with K_i_ values of 197 and 26.1 nM, as shown in Table 3. Compound **7t** without the -OCH_3_ group at the 6-position of *N*-(pyridin-3-yl)quinoline-2-amine is the least potent α-synuclein compound, with a K_i_ value > 500 nM (Table 3), indicating that the -OCH_3_ group plays a critical role for α-synuclein binding activity.

Compounds **7u** and **8a–j** were synthesized using different alkoxy and aryl methoxyl substitutions to replace the 2-fluoroethoxy group in compound **7j**. The in vitro binding data showed that compound **7u** had a moderate potency, with a K_i_ value of 44.6 nM. The K_i_ values of compounds **8a–h** ranged from 16.6 to over 500 nM (Table 3). Compound **8i** containing the ICH=CHCH_2_O- group had a K_i_ value of 5.0 nM. Compound **8j** with two pyridinyl groups introduced at the 2- and 6-positions of the *N*-quinolinyl-2-amine lost its α-synuclein activity, with **a** K_i_ value > 1000 nM.

#### 3.3.2. Structure–Activity Relationship Analysis

To gain additional insight into the binding site of these molecules, a photoaffinity variant (**TZ-CLX**) was synthesized containing a diazirine as a photo-crosslinking motif. Closely resembling the 6-substituted lead compounds **7j** and **8i**, this compound covalently inserts into C-H bonds on nearby residues in the α-synuclein fibril binding site upon irradiation with ultraviolet (UV) light. Using established procedures, the crosslinked fibrils can then be disaggregated, digested using either GluC or trypsin, and analyzed via LC-MS/MS to ascertain residue-level binding information [36]. Rewardingly, **TZ-CLX** exhibited significant selectivity for the C-terminal binding sites of α-synuclein fibrils, with the most abundant peptide identified being AYEMPSEE (residues 124–131, proximal to site 9, Figure 3). Mass adducts were positively identified corresponding to both glutamates 126 and 130, with 130 exhibiting the most labeling. Minor crosslinks were also identified in the N-terminal binding site, primarily via trypsin digestion and corresponding to the peptide EGVVAAAEK (residues 13–21, proximal to site 2, Figure 3). These studies show more significant binding interactions to the N- and C-terminus of α-synuclein fibrils for these quinolinyl scaffolds than we observed for phenyl isoxazole compounds, which are primarily crosslinked to the fibril core. This trend was also observed recently in crosslinking studies involving polyaromatic pyridinyl scaffolds [37], indicating that these more extended systems pick up new sets of interactions and sit in the binding sites differently. These two sets of binding interactions are important to consider when interpreting binding data; competition with [^3^H]BF2286 is primarily expected to involve site 9 binding, whereas the somewhat higher affinities observed for direct radioligand binding and the two sites observed in PD tissue binding may reflect binding to both site 2 and site 9.

#### 3.3.3. Determination of Binding Activity in Human Brain AD Tissues Using [^3^H]PiB

Most potent α-synuclein compounds also bind to Aβ or tau proteins; thus, they lack the selective binding for aggregated α-synuclein versus Aβ and tau proteins, preventing their use as selective α-synuclein PET radiotracers. Both Aβ and tau proteins are highly aggregated in the brains of AD patients, and the availability of AD brain tissues is much better than the availability of PD brain tissues. To determine the selectivity of compound binding toward α-synuclein versus Aβ proteins, the binding potency of the new synthesized compounds toward AD brain tissues were determined using [^3^H]PiB, a well-known Aβ radiotracer. The binding potency of these new compounds was categorized into three ranges, 1 µM, 100 nM, and 10 nM, as shown in Figure 4.

Most of these new compounds had very weak binding vs. [^3^H]PiB in AD tissues in the µM range, suggesting that these new quinolinyl pyridine amines have selective binding with aggregated α-synuclein and not Aβ.

After radiotracers [^11^C]**7f**, [^18^F]**7j**, and [^11^C]**8i** were synthesized for NHP imaging, a direct radioligand binding assay was carried out. The binding affinities (K_d_ values) of compounds [^11^C]**7f** and [^18^F]**7j** were 15.5 ± 4.8 nM and 21 ± 8.7 nM, respectively, towards LBD-amplified fibrils, while [^18^F]**7j** showed a K_d_ value of 19.5 ± 15.5 nM towards PD tissue. These binding affinity data are comparable with the results using [^3^H]BF2286 radioactive competitive binding assays, which showed K_i_ values of 43 and 18 nM towards α-synuclein fibrils, respectively. The binding affinity of [^11^C]**7f** towards AD tissue was 62.7 ± 68.8 nM, with shallow displacement curves. Radiotracer [^18^F]**7j** showed no binding with AD tissues. Thus, these quinoline analogs showed selective binding to α-synuclein fibrils over AD tissues, which is an important step forward given that amyloid-beta pathology frequently exists alongside Lewy body pathology in PD and DLB. In addition, the young control tissue did not show any specific binding in either radiotracer, indicating minimal to no off-target binding of these radiotracers (Figure 5a,b).

As the [^3^H]BF2846 assay showed that the iodine-containing compound **8i** (**TZ61-84**) is very potent for α-synuclein, with a K_i_ value of 5.0 nM, we synthesized [^125^I]**8i** and custom-synthesized [^3^H]**8i**. The in vitro characterizations of [^125^I]**8i** and [^3^H]**8i** binding toward α-synuclein fibrils, PD, AD, and control brain tissues are shown in Figure 6b,c. These data indicated that (a) [^125^I]**8i** binds to α-synuclein-amplified LBD fibrils, with a K_d_ value of 4.8 nM; (b) [^125^I]**8i** is very potent for PD tissues but not for control and AD brain tissues; (c) [^125^I]**8i** has two binding sites with PD tissues, with the high binding site showing K_d_ = 0.61 nM and B_max_ = 50 nM, and the low binding site showing K_d_ = 204 nM and B_max_ = 12 μM; (d) the bottom panel for tritium-labeled [^3^H]**8i** shows K_i_ = 5.4 nM, with the high binding site K_d_ = 1.4 nM and B_max_ = 385 nM and the low binding site K_d_ = 204 nM and B_max_ = 9.1 μM, showing high potency for PD brain tissues and not for AD or control brain tissues.

As shown in Figure 4, we checked whether our compounds bound to α-synuclein fibrils or Aβ protein in AD tissues. Directly using healthy human brain tissues without α-synuclein or Aβ proteins to check a compound’s binding potency can directly confirm the compounds binding specificity. However, for the general screening of new synthesized compounds, the limited available amount of healthy control brain tissues has prevented us from completing such a study. However, if a more promising radiotracer is selected and its tritiated radioligand is made, it could be characterized for young healthy control brain tissues and AD and PD brain tissues, confirming its binding specificity for alpha-synuclein in PD.

### 3.4. Radiochemistry

The in vitro binding data with α-synuclein fibrils identified compounds **7f**, **7j**, and **8i** as ligands that exhibited high binding potency (Table 3).

The potent compound **8**i was labelled with ^125^I, as shown in Figure 4, to provide [^125^I]**8i**, for further in vitro characterization of its binding properties for α-synuclein. Briefly, the tributylstannane precursor **7x** was added into a reaction vial containing 50 µL of freshly prepared 5% sodium acetate in glacial acetic acid, followed by adding 10 µL of [^125^I]NaI and 50 µL of freshly prepared hydrogen peroxide/acetic acid. The reaction mixture was stirred at room temperature for 15 min, and then the radioactive product [^125^I]**8i** was purified using a reverse-phase HPLC analytical C18 column, pushed through a Sep-Pak C18 cartridge for solid-phase extraction (SPE); the radioactive product was eluted in 2 mL of ethanol, ready for the in vitro characterization of its binding toward α-synuclein, Aβ, and tau fibrils, as well as post mortem tissues of PD, AD, and controls.

Three ligands **7f**, **7j**, and **8i** with high binding potency for α-synuclein fibrils, as shown in Table 3, were selected for PET imaging in NHPs; the ^11^C-labeled [^11^C]**7f,** [^11^C]**8i,** and ^18^F-labeled [^18^f]**7j** were radiosynthesized to further test their capability in penetrating the BBB and to determine their pharmacokinetics in NHPs. To synthesize [^11^C]**8i**, the precursor **10** was made following Figure 5. Briefly, the demethylation of compound **7e** using HBr to generate the dihydroxyl compound **9**, followed by selective O-alkylation using (*E*)-3-bromo-1-iodoprop-1-ene, afforded the hydroxyl-pyridinyl-containing precursor **10**. The radiotracer [^11^C]**8i** was made via the O-[^11^C]methylation of **10** using [^11^C]CH_3_OTf in the presence of an aqueous potassium hydroxide solution (5.0 M) in DMF, with a radiochemical yield of ~30% and molar activity > 74 GBq/μmol, decay-corrected to the end of synthesis (EOS). The ^18^F-labeled radiotracer [^18^F]**7j** was made using a two-step procedure. The radioactive intermediate 2-[^18^F]fluoroethyl tosylate was made first, followed by O-alkylation of the –OH group in the precursor **7e** to afford [^18^F]**7j**, with a radiochemistry yield of 25 ± 5% and molar activity > 51 GBq/µmol, decay-corrected to the EOS. The synthesis of these three radiotracers [^11^C]**7f**, [^11^C]**8i,** and [^18^F]**7j** was accomplished smoothly and without any challenges to generate sufficient doses (5–20 mCi) of each radiotracer for NHP PET scans.

### 3.5. PET Brain Imaging Studies in Macaques

To further investigate the pharmacokinetics of [^11^C]**7f**, [^11^C]**8i**, and [^18^F]**7j** for neuroimaging, PET brain scans were performed in three normal healthy cynomolgus macaques using a Siemens microPET Focus 220 scanner. [^11^C]**8i** penetrated the BBB very well, and the initial brain uptake reached standardized uptake values (SUVs) of about 3 g/mL at 3 min post-injection, although the washout kinetics from the brain were slow, with an uptake ratio of about 1.5 from 3 min to 120 min, preventing further evaluation for clinical use (Figure 7c). As shown in Figure 7a,b, [^11^C]**7f** and [^18^F]**7j** had high initial brain uptake rates, which peaked at 3 min post-injection, followed by rapid washout from all brain regions. The NHP PET study data suggested that both [^11^C]**7f** and [^18^F]**7j** penetrated the BBB well and displayed rapid washout pharmacokinetics in the NHP brain. The DVR images show high uptake of these radiotracers in the superior frontal gyrus, caudate, and putamen, which washed out rapidly. In general, both radiotracers showed homogeneous distributions in the brain, which was expected because normal animals do not have aggregated α-synuclein accumulation in any brain region. Further PET brain imaging studies of [^11^C]**7f** and [^18^F]**7j** in animal models containing brain α-synuclein aggregates could demonstrate in vivo target binding to α-synuclein for these radiotracers. The high brain uptake and rapid brain washout pharmacokinetics of both [^11^C]**7f** and [^18^F]**7j** are encouraging. Compared to [^11^C]**7f** and [^18^F]**7j**, which demonstrated both high initial brain uptake and rapid clearance rates, [^11^C]**8i** exhibited relatively slower washout kinetics in NHP PET studies despite its favorable brain entry. This behavior is likely associated with its higher lipophilicity (cLogD = 5.8), which may enhance its non-specific binding and reduce clearance efficiency from non-target brain regions. However, [^11^C]**8i** also demonstrated high binding affinity for α-synuclein fibrils (K_d_ = 5.0 nM) and potent selectivity for PD brain tissues over AD and control tissues, potentially contributing to prolonged retention in target-rich areas. The short half-life of carbon-11 (20.4 min) offers some mitigation of the radiation burden associated with slow washout, allowing useful imaging contrast within the first hour post-injection. Nonetheless, the combination of high affinity and lipophilicity suggests that further optimization to balance target engagement with more favorable pharmacokinetics is warranted.

For this exploratory study, we performed 0–120 min dynamic PET scans in NHPs (*n* = 1 per tracer) to check whether the radiotracers can enter the brain and have suitable washout pharmacokinetics. The data demonstrated that all three radiotracers had good BBB penetration and acceptable brain kinetics. As NHP dynamic scans are very complex and each animal needs at least four weeks of recovery after scanning, it is very challenging to perform more repeated scans at this stage. This design is commonly used for first-in-primate screening studies to identify promising tracers for further evaluation. While logP measurements, full pharmacokinetic modeling, radiometabolite analyses, and efflux transporter evaluations are important for comprehensive tracer characterization, these experiments are typically conducted after the initial screening phase. Due to the complexity of NHP studies and limited availability of resources, these additional evaluations were not included at this stage. We believe the current data are sufficient and provide meaningful evidence for identifying promising radiotracers. Further studies with larger sample sizes and more detailed characterization are planned for the most promising compounds as part of future optimization efforts.

## 4. Conclusions

In this study, *N*-(6-methoxypyridin-3-yl)quinolin-2-amine was identified as a potent pharmacophore for α-synuclein fibrils. The lead structure was optimized with different strategies to synthesize 43 new compounds. The in vitro screening of these new analogues using α-synuclein fibrils and a competitive binding assay using [^3^H]BF2846 as the radioligand showed that several lead compounds have high potency for α-synuclein fibrils and good selectivity. Three lead PET radiotracers [^11^C]**7f**, [^11^C]**8i**, and [^18^F]**7j** were synthesized with good radiochemistry yields and high molar activities. NHP PET brain studies demonstrated that each of these three PET radiotracers was able to penetrate the BBB and enter the brain with favorable pharmacokinetic washout rates. A further evaluation of these new radiotracers may lead to the identification of a promising radiotracer for imaging α-synuclein in vivo. Structural and biological analyses of these new analogues, coupled with the further exploration of diverse structures could help guide the development of new PET tracers and identify a suitable PET tracer for imaging α-synuclein in PD and other related diseases.

## Data Availability

Data will be made available on request.

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
