# Peer review of "Discovery of N-(6-Methoxypyridin-3-yl)quinoline-2-amine Derivatives for Imaging Aggregated α-Synuclein in Parkinson’s Disease with Positron Emission Tomography"

_cells, 2025, doi:10.3390/cells14141108_

Round 1

Reviewer 1 Report

Comments and Suggestions for Authors

The authors reported a new class of N-(6-methoxypyridin-3-yl) quinoline-98-2-amine selective for α-synuclein. They identified three lead tracers labeled with carbon-11 and fluorine-18 and evaluated in non-human primates. The PET NHP brain imaging studies indicated that these tracers had good brain uptake and appropriately fast washout pharmacokinetics. Further optimization and evaluation of this new class of compounds could lead to the development a radiotracer suitable for in vivo imaging of α-synuclein. This is an interesting paper. Publication is recommended after major revision.

  • The manuscript contains excessive first-person pronouns (e.g., 'we', 'our'). In academic writing, it is generally advisable to avoid the first person to maintain objectivity and enhance the credibility of the work.
  • The placement of references is inconsistent, appearing both before and after full stops. Please be consistent throughout the manuscript.
  • Numerous typographical errors are present in the manuscript and supplementary material. Careful proofreading is recommended.
  • Throughout the manuscript, please be consistent about the designation of hours and degrees C.
  • Abstract: “ A photoaffinity variant, TZ-CLX, structurally related to 7j and 8i, preferentially bound to the C-terminal region of α-synuclein fibrils.”---unclear

Suggested modification  “A photoaffinity variant, TZ-CLX, structurally related to 7j and 8i, demonstrated preferential binding to the C-terminal region of α-synuclein fibrils.

  • Introduction: 2nd paragraph, lines 66-67: This paragraph describes the PET tracers. [125I]IDP-4 is not a PET tracer. Therefore, the following sentence is irrelevant.

“[125I]IDP-4 has higher binding affinity ----------

Combine paragraphs 2 and 3

  • Introduction 4th paragraph, Lines 104-106: “We then synthesized [11C]7f, [18F]7j, [11C]8i, [125I]8i and [3H]8i for NHP PET brain imaging studies and in vitro characterizations of their binding profiles for aggregated α-synuclein proteins.

Break this sentence to separate NHP PET brain imaging studies and in vitro characterizations.

  • Figure 1, line 113: include “other” in between and radiotracers.
  • Materials and methods, line 118: include “and used” between source and without
  • Synthesis of 7e, lines 162-164: Include the eluent for chromatography and characterization data.
  • Synthesis of TZ-CLX: Include characterization data
  • Page 5, lines 183-185: “Three concentrations of tested compounds (10nM, 100nM, and 1μM) and [3H]BF2846 (∼4 nM) were added to 50nM of recombinant α-syn fibrils in a working buffer 184

            (50mM Tris-HCl, 0.01% bovine serum albumin (BSA)).---unclear

Suggested modification: Three concentrations of test compounds (10nM, 100nM, and 1μM) were selected and individually mixed with [3H]BF2846 (∼4 nM). Each mixture was then separately added to 50 nM of recombinant α-syn fibrils in a working buffer (50 mM Tris-HCl, 0.01% bovine serum albumin (BSA)). 

  • Page 5, lines 203-205: Modify as suggested above
  • Page 5, 6, line 217, 275: for reference compounds, in chemistry (2.1.), please mention “syntheses of reference compounds are included in the supporting information.
  • Page 6, line 246: Typo “wit”
  • Page 6, line 248 and line 266: replace reactive with “reaction”
  • Page 6, line 257 and throughout the manuscript: Are the yields decay corrected?
  • Page 7, Radiosynthesis: Please include a sentence with a relevant reference about the synthesis of [18F]12.
  • Page 7, lines 302-305: modify the sentence as

 “-----cartridge, washed with 20 mL of water and eluted-----”

  • Page 8, line 349: “ As described below---” replaced with “as described in Figure 2”
  • Page 8, Figure 2: Remove the arrow. Mention on top “literature” and “current work”
  • Page 8, line 357: Replace “modification” with “alteration of the position”
  • Page 9, line 379, scheme 2 and other places: Typo “fluor”
  • Page 9, lines 381-384: Please include reference.
  • Page 9, line 387: replace “realized” with “synthesized”
  • Page 17, lines 569-578: What was the rationale behind the preparation of [¹²⁵I]8i and [³H]8i? Additionally, why were binding affinity and selectivity studies not performed with [¹¹C]8i, as were done with[¹¹C]7f and [¹⁸F]7j?

Supplementary material

General information, Page 3: “All reagents and starting materials used in this manuscript were obtained from ----”

Please remove this sentence, as the information is provided in the Materials section.

Insert “and used” between sources and without. Remove “any”

Page 4, 10, and throughout: correct typo in the sentence “the reaction mixture was filtrated through---” with “filtered”.

Page 10, correct typo “mml” with “mmol”

Page 18, TZ90-23: change “reflux” to “refluxed” and “washing by” to “washed with” 

Page 21: “Three concentrations of tested compounds (10nM, 100nM, and 1μM) and [3H]BF2846 (∼4 nM) were added to 50nM of recombinant α-syn fibrils in a working buffer (50mM Tris-HCl, 0.01% bovine serum albumin (BSA)).”-----unclear.

As mentioned before, suggested modification: Three concentrations of test compounds (10nM, 100nM, and 1μM) were selected and individually mixed with [3H]BF2846 (∼4 nM). Each mixture was then separately added to 50 nM of recombinant α-syn fibrils in a working buffer (50 mM Tris-HCl, 0.01% bovine serum albumin (BSA)). 

Page 22: “Three concentrations of tested compounds (10nM,100nM, and 1μM), [3H]PiB (∼32 nM), and 0.5μg/μL of AD tissue homogenate (Tissue ID: 05-215) in Dulbecco’s phosphate-buffered saline (DPBS) were added to each well of a nonbinding 96 well plate.”---------same as mentioned above

Page 22, c) Direct radioactive---: “. In brief, the standard homologous reference compound was diluted in 30 mM Tris-HCl pH 7.4, 0.1% BSA. Reactions were incubated at 37 ºC for 1 h before quantifying bound radioligand.”

The reaction being referred to here is unclear. Please clarify   

Page 22, section “Direct radioactive competitive binding assay”: “Filters containing the bound ligand were mixed-------"

Is it “filtrate”

Page 25: The picture is upside down.

Page 29-31: The composition of the HPLC buffer in the experimental section (page 29) does not match with the figure (page 30 and 31).

Page 31: What is the radiation peak at ~6 min. Please consider the integration of this peak for the calculation of the radiochemical purity.

Page 32: Mention the reference for “[18F]12”

Page 42: “to reduce salivary gland and respiration ----”,  replace “respiration” with “respiratory”

Page 42: correct the typo of “nornalizations”

Author Response

Reviewer 1:

The authors reported a new class of N-(6-methoxypyridin-3-yl) quinoline-98-2-amine selective for α-synuclein. They identified three lead tracers labeled with carbon-11 and fluorine-18 and evaluated in non-human primates. The PET NHP brain imaging studies indicated that these tracers had good brain uptake and appropriately fast washout pharmacokinetics. Further optimization and evaluation of this new class of compounds could lead to the development a radiotracer suitable for in vivo imaging of α-synuclein. This is an interesting paper.

Thank reviewer 1 for recognizing the importance of our research work.

Q1:  The manuscript contains excessive first-person pronouns (e.g., 'we', 'our'). In academic writing, it is generally advisable to avoid the first person to maintain objectivity and enhance the credibility of the work.

A1: We have made some the revisions.

Q2: The placement of references is inconsistent, appearing both before and after full stops. Please be consistent throughout the manuscript.

A2: We have made the revisions.

Q3: Numerous typographical errors are present in the manuscript and supplementary material. Careful proofreading is recommended.

A3: We have made the revisions.

Q4: Throughout the manuscript, please be consistent about the designation of hours and degrees C.

A4: We have made the revisions.

Q5: Abstract: “A photoaffinity variant, TZ-CLX, structurally related to 7j and 8i, preferentially bound to the C-terminal region of α-synuclein fibrils.” ---unclear; Suggested modification “A photoaffinity variant, TZ-CLX, structurally related to 7j and 8i, demonstrated preferential binding to the C-terminal region of α-synuclein fibrils.

A5: We have made the revisions.

Q6: Introduction: 2nd paragraph, lines 66-67: This paragraph describes the PET tracers. [125I]IDP-4 is not a PET tracer. Therefore, the following sentence is irrelevant. [125I]IDP-4 has higher binding affinity ----------

A6: We have deleted the [125I]IDP-4.

Q7: Combine paragraphs 2 and 3

A7: We have made the revisions.

Q8: Introduction 4th paragraph, Lines 104-106: “We then synthesized [11C]7f, [18F]7j, [11C]8i, [125I]8i and [3H]8i for NHP PET brain imaging studies and in vitro characterizations of their binding profiles for aggregated α-synuclein proteins. Break this sentence to separate NHP PET brain imaging studies and in vitro characterizations.

A8: We have made the revisions.

Q9: Figure 1, line 113: include “other” in between and radiotracers.

A9: We have made the revisions.

Q10: Materials and methods, line 118: include “and used” between source and without

A10: We have made the revisions.

Q11: Synthesis of 7e, lines 162-164: Include the eluent for chromatography and characterization data.

A11: This information was provided in the supporting information.

Q12: Synthesis of TZ-CLX: Include characterization data

A12: The synthesis and characterization of TZ-CLX was provided in the supporting information ‘5. TZ-CLX Synthesis and Characterization’.

Q13: Page 5, lines 183-185: “Three concentrations of tested compounds (10nM, 100nM, and 1μM) and [3H]BF2846 (4 nM) were added to 50nM of recombinant α-syn fibrils in a working buffer 184

            (50mM Tris-HCl, 0.01% bovine serum albumin (BSA)). ---unclear

Suggested modification: Three concentrations of test compounds (10nM, 100nM, and 1μM) were selected and individually mixed with [3H]BF2846 (4 nM). Each mixture was then separately added to 50 nM of recombinant α-syn fibrils in a working buffer (50 mM Tris-HCl, 0.01% bovine serum albumin (BSA)).

A13: We have made the revisions.

Q14: Page 5, lines 203-205: Modify as suggested above

A14: We have made the revisions.

Q15: Page 5, 6, line 217, 275: for reference compounds, in chemistry (2.1.), please mention “syntheses of reference compounds are included in the supporting information.

A15: We have made the revisions.

Q16: Page 6, line 246: Typo “wit”

A16: We have made the revisions.

Q17: Page 6, line 248 and line 266: replace reactive with “reaction”

A17: We have made the revisions.

Q18: Page 6, line 257 and throughout the manuscript: Are the yields decay corrected?

A18: Yes, they are.

Q19: Page 7, Radiosynthesis: Please include a sentence with a relevant reference about the synthesis of [18F]12.

A19: We have added the reference.

Q20: Page 7, lines 302-305: modify the sentence as “-----cartridge, washed with 20 mL of water and eluted-----”

A20: We have made the revisions.

Q21: Page 8, line 349: “As described below---” replaced with “as described in Figure 2”

A21: We have made the revisions.

Q22: Page 8, Figure 2: Remove the arrow. Mention on top “literature” and “current work”

A22: We have made the revisions.

Q23: Page 8, line 357: Replace “modification” with “alteration of the position”

A23: We have made the revisions.

Q24: Page 9, line 379, scheme 2 and other places: Typo “fluor”

A24: We have made the revisions.

Q25: Page 9, lines 381-384: Please include reference.

A25: We have added the reference.

Q26: Page 9, line 387: replace “realized” with “synthesized”

A26: We have made the revisions.

Q27: Page 17, lines 569-578: What was the rationale behind the preparation of [¹²⁵I]8i and [³H]8i? Additionally, why were binding affinity and selectivity studies not performed with [¹¹C]8i, as were done with[¹¹C]7f and [¹⁸F]7j?

A27: Thank you for this thoughtful question. The rationale for preparing [¹²⁵I]8i and [³H]8i was to enable comprehensive in vitro characterization of compound 8i, including high-sensitivity binding affinity and selectivity assays. These isotopes, with their longer half-lives, are well suited for saturation binding, competition assays, and autoradiography, which require extended incubation times and accurate quantification. In contrast, [¹¹C]8i was specifically synthesized for in vivo PET imaging. Due to the short half-life of carbon-11 (~20 min), it is suboptimal for in vitro studies that require prolonged handling or equilibrium binding conditions. While [¹¹C]7f and [¹⁸F]7j underwent in vitro displacement assays during early-stage evaluation to confirm their binding behavior, we subsequently optimized our workflow by utilizing [¹²⁵I]- and [³H]-labeled versions of 8i to obtain more stable and reproducible vitro binding data.

Q28: Supplementary material

Q29: General information, Page 3: “All reagents and starting materials used in this manuscript were obtained from ----”

Please remove this sentence, as the information is provided in the Materials section.

Insert “and used” between sources and without. Remove “any”

A29We have removed this sentence.

Q30: Page 4, 10, and throughout: correct typo in the sentence “the reaction mixture was filtrated through---” with “filtered”.

A30We have corrected the typographical error.

Q31: Page 10, correct typo “mml” with “mmol”

A31: We have corrected the typographical error.

Q32: Page 18, TZ90-23: change “reflux” to “refluxed” and “washing by” to “washed with” 

A32: We have corrected the grammatical error.

Q33: Page 21: “Three concentrations of tested compounds (10nM, 100nM, and 1μM) and [3H]BF2846 (4 nM) were added to 50nM of recombinant α-syn fibrils in a working buffer (50mM Tris-HCl, 0.01% bovine serum albumin (BSA)).” -----unclear.

As mentioned before, suggested modification: Three concentrations of test compounds (10nM, 100nM, and 1μM) were selected and individually mixed with [3H]BF2846 (4 nM). Each mixture was then separately added to 50 nM of recombinant α-syn fibrils in a working buffer (50 mM Tris-HCl, 0.01% bovine serum albumin (BSA)). 

A33: We have revised the wording and adopted the suggested phrasing.

Q34: Page 22: “Three concentrations of tested compounds (10nM,100nM, and 1μM), [3H]PiB (32 nM), and 0.5μg/μL of AD tissue homogenate (Tissue ID: 05-215) in Dulbecco’s phosphate-buffered saline (DPBS) were added to each well of a nonbinding 96 well plate.” ---------same as mentioned above

A34: We have revised the wording and adopted the suggested phrasing.

Q35: Page 22, c) Direct radioactive---: “. In brief, the standard homologous reference compound was diluted in 30 mM Tris-HCl pH 7.4, 0.1% BSA. Reactions were incubated at 37 ºC for 1 h before quantifying bound radioligand.”

The reaction being referred to here is unclear. Please clarify

A35: We have updated the description and provided more details of the direct binding assay.

Q36: Page 22, section “Direct radioactive competitive binding assay”: “Filters containing the bound ligand were mixed-------"

Is it “filtrate”

A36: We have corrected the typographical error.

Q37: Page 25: The picture is upside down.

A37: We have adjusted the orientation of the picture.

Q38: Page 29-31: The composition of the HPLC buffer in the experimental section (page 29) does not match with the figure (page 30 and 31).

A38: We have made the revisions.

Q39: Page 31: What is the radiation peak at ~6 min. Please consider the integration of this peak for the calculation of the radiochemical purity.

A39: The small peak at ~6 min accounts for less than 5% of total radioactivity and likely represents a minor byproduct. Radiochemical purity across multiple syntheses consistently exceeded 99%. The shown chromatogram is from an earlier run; while we are unable to retrieve the original raw file due to lab renovation, the consistent high purity observed in these batches supports the validity of this result.

Q40: Page 32: Mention the reference for “[18F]12”

A40: We have added the reference for [18F]12.

Q41: Page 42: “to reduce salivary gland and respiration ----”, replace “respiration” with “respiratory”

A41: We have made the revisions.

Q42: Page 42: correct the typo of “nornalizations” 

A42: We have made the revisions.

Reviewer 2 Report

Comments and Suggestions for Authors

The manuscript entitled “Discovery of N-(6-Methoxypyridin-3-yl)quinoline-2-amine Derivatives for Imaging Aggregated α-Synuclein in Parkinson’s Disease with Positron Emission Tomography” submitted by Zhao Haiyang et al. describes the derivatives of N-(6-Methoxypyridin-3-yl)quinoline-2-amine for the exploration of α-synuclein in PD (in vitro) and healthy in vivo PET imaging. Please see comments below:
1)    For section 2.3 and radiochemistry procedures, please include details such as volume of NaOH in all the procedures where necessary. For example, line 245 and page 25 of SI document.
2)    Scheme 2 need to be better organized to follow compounds numbers sequentially. Please update.
3)    There are several typos throughout the manuscript, e.g line 398 and 400, please check the entire manuscript.
4)    For fig 4 and 5, please include control tissues for comparison with α-Synuclein fibrils and Aβ in AD tissues.
5)    For fig. 5, please label the figures with 5a and 5b panels and seems [18F]7j structure is missing, please revise.
6)    For fig 6, seems the authors have used two types of numbering, for example [125I]TZ61-84 and [125I]8i, both refers the same compound. Could you please make a consistent labeling to follow throughout the manuscript and SI.
7)    For fig 6, please include direct binding of [11C]8i with banner PD, AD and control tissues as the authors have demonstrated a bit variability with [125I] and [3H]-labeled TZ61-84 or 8i?
8)    For scheme 4, step a need all the necessary reagents to be shown. Please correct.
9)    For scheme 5b, the compound number looks to be different when compared to the scheme.
10)    For table 1-3, could you please include experimentally determined LogP values to better predict their lipophilicity and include half-lives (t1/2) of at least [11C]7f, [11C]8i, [11C]7j in NHPs to better understand their washout periods.
11)    For fig 7, have the authors have normalized to blood? How does the authors ensure that the tracers crossing the BBB in early time points versus later time points? Would it be feasible to derive the brain uptake if it is less than SUV of less than 2 of ideal BBB penetrant tracer?
12)    What would be the influence of efflux transporters on uptake of [11C]8i?
13)     Have the authors tested the stability of radiotracers [11C]7f, [11C]8i, [11C]7j in normal and PD or AD tissues over time? or in the serum over 2 hr time at least?
14)     For fig 7d, what time points do the images represent at? Please include the timepoint in the label or on the image.
15)     For fig 7, it looks like authors have used n=1 (one animal per tracer) for PET imaging, however, n=3 (minimum) is highly recommended to determine the statistical differences, if any. Please revise and update.
16)     In the discussion section, please include the discussion of how or potential reasons of [11C]8i being high uptake despite being high cLogD and potent for PD tissues (in vitro) and how the half-life or other PK properties helps mitigate high uptake and slow wash out? 

Please consider as major revision and update.

Author Response

Reviewer 2:

Comments and Suggestions for Authors

The manuscript entitled “Discovery of N-(6-Methoxypyridin-3-yl)quinoline-2-amine Derivatives for Imaging Aggregated α-Synuclein in Parkinson’s Disease with Positron Emission Tomography” submitted by Zhao Haiyang et al. describes the derivatives of N-(6-Methoxypyridin-3-yl)quinoline-2-amine for the exploration of α-synuclein in PD (in vitro) and healthy in vivo PET imaging. Please see comments below:

Thank reviewer 2 for recognizing the importance of our research work.

Q1: For section 2.3 and radiochemistry procedures, please include details such as volume of NaOH in all the procedures where necessary. For example, line 245 and page 25 of SI document.

A1: We have made the revisions.

Q2: Scheme 2 need to be better organized to follow compounds numbers sequentially. Please update.

A2: We have made the revisions.

Q3: There are several typos throughout the manuscript, e.g line 398 and 400, please check the entire manuscript.

A3: We have made the revisions.

Q4: For fig 4 and 5, please include control tissues for comparison with α-Synuclein fibrils and Aβ in AD tissues.
A4: We thank the reviewer for the comment. As described in our manuscript, for our experiment in Fig 4, the goal is to determine the selectivity of compound binding toward α-synuclein versus Aβ protein and identify candidate compounds that are more specific to α-synuclein. The binding in normal control tissues is not included and considered at this stage. After we selected the candidate compounds that were more specificity to α-synuclein, we then moved forward and determined the binding properties in PD, AD, healthy control and other tissues as shown in Figure 5 and 6. In Figures 5 and 6, we have included binding data for α-synuclein radioligands in PD, AD, and young healthy control brain tissues. These data demonstrate strong binding in PD tissues and no binding in both AD and control tissues, confirming the selectivity of our radiotracers for α-synuclein over Aβ and nonspecific targets.

Q5: For fig. 5, please label the figures with 5a and 5b panels and seems [18F]7j structure is missing, please revise.

A5: We have made the revisions.

Q6: For fig 6, seems the authors have used two types of numbering, for example [125I]TZ61-84 and [125I]8i, both refers the same compound. Could you please make a consistent labeling to follow throughout the manuscript and SI.

A6: We have made the revisions.

Q7: For fig 6, please include direct binding of [11C]8i with banner PD, AD and control tissues as the authors have demonstrated a bit variability with [125I] and [3H]-labeled TZ61-84 or 8i?

A7: We appreciate the reviewer’s suggestion. Due to the short half-life of carbon-11 (~20 min), autoradiography with [¹¹C]8i on human brain sections is technically impractical. Instead, we used [¹²⁵I]- and [³H]-labeled analogs to assess tissue binding, which offer better stability and sensitivity for in vitro studies. The observed variability reflects biological differences rather than tracer limitations. While future [¹¹C]8i autoradiography may be considered, current data sufficiently validate target binding.

Q8: For scheme 4, step a need all the necessary reagents to be shown. Please correct.

A8: We have made the revisions.

Q9: For scheme 5b, the compound number looks to be different when compared to the scheme.

A9: We have made the revisions.

Q10: For table 1-3, could you please include experimentally determined LogP values to better predict their lipophilicity and include half-lives (t1/2) of at least [11C]7f, [11C]8i, [11C]7j in NHPs to better understand their washout periods.

A10: We thank the reviewer for this valuable suggestion. Currently, experimental LogP values have not been measured; the lipophilicity of the compounds was predicted using computational methods. We agree that experimentally determined LogP would provide a more accurate understanding of BBB permeability, and we plan to measure these values in future studies. Regarding the biological half-lives (t₁/₂) of [¹¹C]7f, [¹¹C]8i, and [¹¹C]7j in non-human primates, dynamic PET imaging studies have been performed; however, the exact washout half-lives have not yet been formally calculated. We are currently analyzing the time-activity curves (TACs) from NHP PET scans, and we plan to report the quantitative pharmacokinetic parameters, including t₁/₂, in a follow-up study.

Q11: For fig 7, have the authors have normalized to blood? How does the authors ensure that the tracers crossing the BBB in early time points versus later time points? Would it be feasible to derive the brain uptake if it is less than SUV of less than 2 of ideal BBB penetrant tracer?

A11: We thank the reviewer for the thoughtful questions. We did not normalize to the blood as we did not perform blood input functions for each tracer due to the limitation of resource and availability of monkey studies. We agree that kinetic modeling with blood input function will reflect more accurate brain uptake levels, however, our methods are well accepted generic analysis for tracer uptake and can support our conclusion well that these penetrate the BBB well with high brain uptake. The potential effect from blood may but can only impact the uptake of very first few minutes and thus will not impact our conclusions since all our tracer still have high uptake from several to at least 20 minutes. In Figure 7, the data are presented as standardized uptake values (SUVs) without blood normalization. We evaluated BBB permeability based on early-phase dynamic PET imaging. All three tracers ([¹¹C]7f, [¹¹C]8i, and [¹⁸F]7j) showed rapid brain entry, with SUV values exceeding 2 within the first 3 minutes post-injection, followed by distinct washout kinetics. This pattern supports effective BBB penetration. While SUVR or brain-to-blood ratios were not calculated in this study, the observed dynamic uptake profiles and clearance behavior suggest meaningful CNS exposure. Future studies will include kinetic modeling and blood-based normalization to further characterize tracer pharmacokinetics.

Q12: What would be the influence of efflux transporters on uptake of [11C]8i?

A12: We appreciate the reviewer’s insightful question. At this stage, we have not conducted specific experiments to assess whether [¹¹C]8i is a substrate of efflux transporters such as P-glycoprotein (P-gp). however, further studies using transporter inhibitors in rodent or NHP models are needed to confirm this. We plan to include such investigations in future work to better understand the pharmacokinetics and potential limitations of [¹¹C]8i as a CNS imaging agent.

Q13: Have the authors tested the stability of radiotracers [11C]7f, [11C]8i, [11C]7j in normal and PD or AD tissues over time? or in the serum over 2 hr time at least?

A13: We thank the reviewer for the important question. At present, we have not conducted radiometabolism studies of [¹¹C]7f, [¹¹C]8i, or [¹¹C]7j in brain tissues or plasma. We fully recognize the importance of evaluating in vivo stability and metabolic profiles, and such studies are being prioritized in our ongoing work.

Q14: For fig 7d, what time points do the images represent at? Please include the timepoint in the label or on the image.

A14: We have made the revisions.

Q15: For fig 7, it looks like authors have used n=1 (one animal per tracer) for PET imaging, however, n=3 (minimum) is highly recommended to determine the statistical differences, if any. Please revise and update.

A15: We appreciate the reviewer’s recommendation. As noted, Figure 7 presents PET imaging data from a single nonhuman primate per tracer. These initial studies were intended as pilot evaluations to assess brain uptake, BBB penetration, and pharmacokinetic profiles. Given limitation of resource and availability of NHP imaging studies, early-phase feasibility assessments are commonly performed with n = 1 per tracer. We agree that future studies with larger cohorts (n ≥ 3) will be essential to determine statistical differences and improve confidence in tracer performance. Such studies are currently being planned to support further preclinical validation.

Q16: In the discussion section, please include the discussion of how or potential reasons of [11C]8i being high uptake despite being high cLogD and potent for PD tissues (in vitro) and how the half-life or other PK properties helps mitigate high uptake and slow wash out? 

A16: Thank you for the helpful suggestion. We have added a paragraph at the end of the Discussion section (page 19, line 679-691) to address this point. We discussed possible reasons for the slower washout of [¹¹C]8i, including its high cLogD and strong binding to PD tissues, and how the short half-life of carbon-11 may help mitigate slow clearance.

Reviewer 3 Report

Comments and Suggestions for Authors

Dear Authors,

Thank you for your work.

The authors developed new radiotracers for PET imaging of synuclein in Parkinson's disease (PD) and evaluated them in vitro and in vivo. They determined the binding specificity and selectivity toward synuclein in AD and PD samples. The authors even identified two different binding sites of TZ-CLX, a photoaffinity variant with a structure very similar to that of two of the investigated imaging probes. The authors present different radiolabeling methods that include C-11, F-18, I-125, and H-3 as radionuclides. PET imaging in non-human primates (NHPs) revealed good brain uptake, with differences in the washout phase. However, the background clearance is fast enough to potentially provide good signal-to-noise ratio (SNR) in future studies. Further studies should address whether the background signal is low enough to provide sufficient contrast.

The manuscript is clear and understandable, well-structured, and the methods and results are well-presented. There are no ethical concerns, and the references are well-selected.

Minor:

Please add the MR scanner used.

Kind regards and congrats

Author Response

Reviewer 3:

The authors developed new radiotracers for PET imaging of synuclein in Parkinson's disease (PD) and evaluated them in vitro and in vivo. They determined the binding specificity and selectivity toward synuclein in AD and PD samples. The authors even identified two different binding sites of TZ-CLX, a photoaffinity variant with a structure very similar to that of two of the investigated imaging probes. The authors present different radiolabeling methods that include C-11, F-18, I-125, and H-3 as radionuclides. PET imaging in non-human primates (NHPs) revealed good brain uptake, with differences in the washout phase. However, the background clearance is fast enough to potentially provide good signal-to-noise ratio (SNR) in future studies. Further studies should address whether the background signal is low enough to provide sufficient contrast.

The manuscript is clear and understandable, well-structured, and the methods and results are well-presented. There are no ethical concerns, and the references are well-selected.

Thank reviewer 3 for recognizing the importance of our research work.

Q1: Please add the MR scanner used.

A1: MR data were acquired prior to PET image acquisition with a Siemens 3T Trio scanner, with an extremity head coil in the coronal direction and the following spin echo sequence (TE = 3.34 ms, TR = 2530 ms, fip angle = 7°, thickness = 0.50 mm, feld-of-view = 140 mm, image matrix = 256 × 256 × 176, voxel size = 0.547 × 0.547 × 0.500 mm). We have incorporated this information on page 42 of the Supporting Information.

Reviewer 4 Report

Comments and Suggestions for Authors

Dr. Haiyang Zhao et al. report about the novel candidates of PET probes for imaging aggregated a-synuclein. This manuscript is very interesting and promising, however there are several minor comments on this as follows.

  1. In Abstract, authors suggest that TZ-CLX bind to the C-terminal region of α-synuclein fibrils. Is TZ-CLX stacked among the b-sheet structure of α-synuclein as observed between [11C]PiB and amyloid-beta?
  2. In Abstract, authors firstly describe about the in vivo kinetic analyses of [11C]7f, [18F]7j, and [11C]8i in the brain of nonhuman primate using PET, followed by about the in vitro binding analysis of [125I]8i to assess its specificity and affinity. The order of these issues should be reversed, because researchers usually apply novel PET probes for PET imaging after the confirmation of the properties such as their specificity and affinity for the target molecules.
  3. In 1. Introduction, authors describe regarding the specificity of [18F]ACI-12589 for MSA-type α-synuclein. What type of α-synuclein fibrils were used in this study? If they did not use the PD-type α-synuclein fibrils, Title of this manuscript should be changed to exclude the words of “in Parkinson’s Disease”.
  4. Throughout the manuscript, the description of compound names is inconsistent, such as TZ6184, TZ-6184, TZ-61-84, and TZ61-84 and so on. The consistent description should be needed.
  5. In Figure 5a and b, [11C]7f and [18F]7j were displaced by TZ-55-107 and TZ-61-44, respectively. Since 7f is TZ-55-107 and 7j is TZ-61-44, the description of [11C]7f and [18F]7j should be changed to [11C]7f (TZ-55-107) and [18F]7j (TZ-61-44) as done in Figure 7 to show more clearly that these was self-displacement studies.
  6. In Figure 5b, the “Bound” level of [18F]7j to AD tissue seems to be about 60 % of that to PD tissue even at lowest TZ-61-44, revealing its high nonspecific binding and/or low specificity to PD tissue. This point should be discussed whether [18F]7j has actually proper as a PET imaging probe.
  7. In the legend of Figure 6, [125I/3H]8i and [125I/3H]TZ61-84 were mixing. It should be changes as pointed out in comment 5.
  8. In Figure 7, there are no data about the metabolic profiles, which should be one of the most important properties to judge whether these compounds can be applicable for PET imaging.

Author Response

Reviewer 4:

Comments and Suggestions for Authors

Dr. Haiyang Zhao et al. report about the novel candidates of PET probes for imaging aggregated a-synuclein. This manuscript is very interesting and promising.

Thank reviewer 4 for recognizing the importance of our research work.

Q1: In Abstract, authors suggest that TZ-CLX bind to the C-terminal region of α-synuclein fibrils. Is TZ-CLX stacked among the b-sheet structure of α-synuclein as observed between [11C]PiB and amyloid-beta?

A1: α-Synuclein fibril is assembled into well-ordered β-sheets, and yes, TZ-CLX is most likely to interact with these β-sheets structures as [11C]PiB and amyloid-beta. Our data indicate TZ-CLX is selective to C-terminal and N-terminal of α-Synuclein fibril and particularly preferred C-terminals, different from previously reported phenyl isoxazole compounds which primarily recognized the fibril core, and therefore, possibly more specific to α-Synuclein fibril structures. In fact, our binding data show 7j and 8i that are close to the structure of TZ-CLX are selective to PD tissues over AD and control tissues.

Q2: In Abstract, authors firstly describe about the in vivo kinetic analyses of [11C]7f, [18F]7j, and [11C]8i in the brain of nonhuman primate using PET, followed by about the in vitro binding analysis of [125I]8i to assess its specificity and affinity. The order of these issues should be reversed, because researchers usually apply novel PET probes for PET imaging after the confirmation of the properties such as their specificity and affinity for the target molecules.

A2: Thank you for your suggestion. While it is standard practice to confirm target affinity prior to in vivo imaging, our tracer development workflow was structured to first evaluate candidate compounds based on their brain uptake and pharmacokinetic profiles in nonhuman primates. This allowed us to prioritize tracers with favorable in vivo performance before committing to detailed in vitro binding studies. Compound 8i demonstrated promising brain penetration and washout kinetics in NHP PET scans, and was therefore selected for further in vitro assessment. As shown in Figure 6a, we used a homologous competition binding assay with [¹²⁵I]8i and unlabeled 8i to estimate its binding affinity to α-synuclein fibrils, yielding an apparent Kd of 4.8 nM. Given that α-synuclein fibrils are structurally heterogeneous and often contain multiple binding domains, and that short-lived PET isotopes such as [¹¹C] are suboptimal for prolonged or high-throughput binding assays, this sequential approach allowed us to balance in vivo feasibility with quantitative target engagement. Accordingly, we have retained the current structure in the Abstract to reflect our actual experimental sequence and practical tracer development considerations.

Q3: In 1. Introduction, authors describe regarding the specificity of [18F]ACI-12589 for MSA-type α-synuclein. What type of α-synuclein fibrils were used in this study? If they did not use the PD-type α-synuclein fibrils, Title of this manuscript should be changed to exclude the words of “in Parkinson’s Disease”

A3: Thank you for your insightful comment. In our study, we used both brain tissues from Parkinson’s disease (PD) patients and α-synuclein fibrils amplified from Lewy body dementia (LBD) patient brain homogenates. Specifically, PD brain tissues were used in autoradiography experiments to directly assess radiotracer binding in human disease-relevant samples (see Figure 5 and Figure 6), while LBD-amplified α-synuclein fibrils were employed in in vitro binding assays for reproducibility and controlled evaluation. Given the well-established structural and pathological similarities between PD and LBD α-synuclein aggregates—particularly the presence of Lewy bodies—LBD-amplified fibrils are widely recognized as a representative model of PD-type α-synuclein. Therefore, we believe the current title appropriately reflects the disease relevance and clinical context of our study.

Q4: Throughout the manuscript, the description of compound names is inconsistent, such as TZ6184, TZ-6184, TZ-61-84, and TZ61-84 and so on. The consistent description should be needed.

A4: We have revised the related description and used TZ61-84 to ensure consistency in the description of compound names.

Q5: In Figure 5a and b, [11C]7f and [18F]7j were displaced by TZ-55-107 and TZ-61-44, respectively. Since 7f is TZ-55-107 and 7j is TZ-61-44, the description of [11C]7f and [18F]7j should be changed to [11C]7f (TZ-55-107) and [18F]7j (TZ-61-44) as done in Figure 7 to show more clearly that these was self-displacement studies.

A5: We have made the revisions.

Q6: In Figure 5b, the “Bound” level of [18F]7j to AD tissue seems to be about 60 % of that to PD tissue even at lowest TZ61-44, revealing its high nonspecific binding and/or low specificity to PD tissue. This point should be discussed whether [18F]7j has actually proper as a PET imaging probe.

A6: The binding of our radioligands to each group of tissue samples were determined by competition binding assay and calculated in Kd (the dissociation constant). We have found low Kd in LBD amplified fibrils and PD tissues, whereas the Kd was high and not able to be determined in AD and control tissues, and thus, support our conclusion that [18F]7j is more specific to PD than AD and control tissues. The ‘Bound’ level was relative level of radioligand at the time of counting and was not comparable among groups.

Q7: In the legend of Figure 6, [125I/3H]8i and [125I/3H]TZ61-84 were mixing. It should be changes as pointed out in comment 5.

A7: We have made the revisions.

Q8: In Figure 7, there are no data about the metabolic profiles, which should be one of the most important properties to judge whether these compounds can be applicable for PET imaging.

A8: The metabolic profiling studies of the tracers are currently underway in our laboratory.

Round 2

Reviewer 1 Report

Comments and Suggestions for Authors

Thank you for revising the manuscript. The responses to the reviewers’ comments are satisfactory, and the revisions have been appropriately made. The manuscript looks good. This reviewer has no additional comments.

Author Response

We sincerely thank the reviewer recognizing our efforts on revision and satisfying our manuscript.

Reviewer 2 Report

Comments and Suggestions for Authors

Q4: For fig 4 and 5, please include control tissues for comparison with α-Synuclein fibrils and Aβ in AD tissues.
A4: We thank the reviewer for the comment. As described in our manuscript, for our experiment in Fig 4, the goal is to determine the selectivity of compound binding toward α-synuclein versus Aβ protein and identify candidate compounds that are more specific to α-synuclein. The binding in normal control tissues is not included and considered at this stage. After we selected the candidate compounds that were more specificity to α-synuclein, we then moved forward and determined the binding properties in PD, AD, healthy control and other tissues as shown in Figure 5 and 6. In Figures 5 and 6, we have included binding data for α-synuclein radioligands in PD, AD, and young healthy control brain tissues. These data demonstrate strong binding in PD tissues and no binding in both AD and control tissues, confirming the selectivity of our radiotracers for α-synuclein over Aβ and nonspecific targets.

R: It would have been great if the authors could have included control tissue to see any non  α-synuclein or non Aβ fibrils as well.
Q7: For fig 6, please include direct binding of [11C]8i with banner PD, AD and control tissues as the authors have demonstrated a bit variability with [125I] and [3H]-labeled TZ61-84 or 8i?
A7: We appreciate the reviewer’s suggestion. Due to the short half-life of carbon-11 (~20 min), autoradiography with [¹¹C]8i on human brain sections is technically impractical. Instead, we used [¹²⁵I]- and [³H]-labeled analogs to assess tissue binding, which offer better stability and sensitivity for in vitro studies. The observed variability reflects biological differences rather than tracer limitations. While future [¹¹C]8i autoradiography may be considered, current data sufficiently validate target binding.

Q10: For table 1-3, could you please include experimentally determined LogP values to better predict their lipophilicity and include half-lives (t1/2) of at least [11C]7f, [11C]8i, [11C]7j in NHPs to better understand their washout periods.
A10: We thank the reviewer for this valuable suggestion. Currently, experimental LogP values have not been measured; the lipophilicity of the compounds was predicted using computational methods. We agree that experimentally determined LogP would provide a more accurate understanding of BBB permeability, and we plan to measure these values in future studies. Regarding the biological half-lives (t₁/₂) of [¹¹C]7f, [¹¹C]8i, and [¹¹C]7j in non-human primates, dynamic PET imaging studies have been performed; however, the exact washout half-lives have not yet been formally calculated. We are currently analyzing the time-activity curves (TACs) from NHP PET scans, and we plan to report the quantitative pharmacokinetic parameters, including t₁/₂, in a follow-up study.

R: It would be ideal to include logP and PK study results as part of this manuscript with n=2 ( 2 NHPs per tracer) with stats.
Q11: For fig 7, have the authors have normalized to blood? How does the authors ensure that the tracers crossing the BBB in early time points versus later time points? Would it be feasible to derive the brain uptake if it is less than SUV of less than 2 of ideal BBB penetrant tracer?
A11: We thank the reviewer for the thoughtful questions. We did not normalize to the blood as we did not perform blood input functions for each tracer due to the limitation of resource and availability of monkey studies. We agree that kinetic modeling with blood input function will reflect more accurate brain uptake levels, however, our methods are well accepted generic analysis for tracer uptake and can support our conclusion well that these penetrate the BBB well with high brain uptake. The potential effect from blood may but can only impact the uptake of very first few minutes and thus will not impact our conclusions since all our tracer still have high uptake from several to at least 20 minutes. In Figure 7, the data are presented as standardized uptake values (SUVs) without blood normalization. We evaluated BBB permeability based on early-phase dynamic PET imaging. All three tracers ([¹¹C]7f, [¹¹C]8i, and [¹⁸F]7j) showed rapid brain entry, with SUV values exceeding 2 within the first 3 minutes post-injection, followed by distinct washout kinetics. This pattern supports effective BBB penetration. While SUVR or brain-to-blood ratios were not calculated in this study, the observed dynamic uptake profiles and clearance behavior suggest meaningful CNS exposure. Future studies will include kinetic modeling and blood-based normalization to further characterize tracer pharmacokinetics.

R: This needs to be addressed with required stats and n=2 or 3 per tracer.
Q12: What would be the influence of efflux transporters on uptake of [11C]8i?
A12: We appreciate the reviewer’s insightful question. At this stage, we have not conducted specific experiments to assess whether [¹¹C]8i is a substrate of efflux transporters such as P-glycoprotein (P-gp). however, further studies using transporter inhibitors in rodent or NHP models are needed to confirm this. We plan to include such investigations in future work to better understand the pharmacokinetics and potential limitations of [¹¹C]8i as a CNS imaging agent.

R: Please explore and furnish the results.

Q13: Have the authors tested the stability of radiotracers [11C]7f, [11C]8i, [11C]7j in normal and PD or AD tissues over time? or in the serum over 2 hr time at least?
A13: We thank the reviewer for the important question. At present, we have not conducted radiometabolism studies of [¹¹C]7f, [¹¹C]8i, or [¹¹C]7j in brain tissues or plasma. We fully recognize the importance of evaluating in vivo stability and metabolic profiles, and such studies are being prioritized in our ongoing work.

R: Please study the stability and report any potential metabolites if any.
Q14: For fig 7d, what time points do the images represent at? Please include the timepoint in the label or on the image.
A14: We have made the revisions.

R: Revised version does not represent the times of the images in Fig 7d.

Q15: For fig 7, it looks like authors have used n=1 (one animal per tracer) for PET imaging, however, n=3 (minimum) is highly recommended to determine the statistical differences, if any. Please revise and update.
A15: We appreciate the reviewer’s recommendation. As noted, Figure 7 presents PET imaging data from a single nonhuman primate per tracer. These initial studies were intended as pilot evaluations to assess brain uptake, BBB penetration, and pharmacokinetic profiles. Given limitation of resource and availability of NHP imaging studies, early-phase feasibility assessments are commonly performed with n = 1 per tracer. We agree that future studies with larger cohorts (n ≥ 3) will be essential to determine statistical differences and improve confidence in tracer performance. Such studies are currently being planned to support further preclinical validation.

R: Please include more animal per tracer and provide stats to be able to predict ideal responses.

Based on the responses provide by the authors, this research work needs additional experiments to determine the suitability of BBB penetration for the investigative compounds. Please perform the required experimentation and resubmit.

Author Response

Q4R: It would have been great if the authors could have included control tissue to see any non α-synuclein or non Aβ fibrils as well.

Q4 follow up: We thank the reviewer for the thoughtful follow-up comment. We agree that including additional control tissues without α-synuclein and Aβ fibrils could provide complementary insight into the ligand binding selectivity. Our data indeed included young healthy control subjects in Figure 5 and 6 for more detailed characterization of our leading compounds. For Figure 4, the experiments focus on determining if α-synuclein potent compounds bind toward Aβ through a screening approach, we don’t think this impacts our decision if a ligand is selective binding toward alpha-synuclein over Aβ fibrils. Figures 5 and 6, clearly demonstrate that compounds have high selectivity for α-synuclein, evidenced by strong binding in PD tissues and minimal to no binding in both AD and young healthy control subjects.

Q10R: It would be ideal to include logP and PK study results as part of this manuscript with n=2 (2 NHPs per tracer) with stats.

Q10 follow-up: We agree with the reviewer including experimental LogP and pharmacokinetics (PK) be ideal. However, for the lead radiotracers, we had completed and presented the two hrs dynamic PET scan, the brain time tissue activity curves already showed whether the radiotracer was able to enter the brain and its brain washout pharmacokinetics. We understand the measurement of log P and PK is also able to predict a radiotracer can enter into the brain and its brain pharmacokinetics. In our future radiosynthesis of radiotracer for the brain, we will include this measurement as evidence of whether a radiotracer can enter the brain. Although we understand the more repeated PET scans we perform, the better we can confirm a radiotracer’s in vivo reproducibility. However, nonhuman primate brain 0-120 min dynamic scan is more complex than multiple scans for rodent brains. After a 0-120 dynamic scan, the monkey was required to take at least 4 weeks for recovery based on our institutional animal use and protection committee request. It is very challenging to have n > 4 for initially check whether a radiotracer can enter the brain and has suitable brain washout pharmacokinetics. However, for most promising radiotracer (s), we will perform multiple scans (n>4) to check their reproducibility for same animal and multiple animals prior to determining further toxicity and dosimetry validations.

Q11R: This needs to be addressed with required stats and n=2 or 3 per tracer.

Q11 follow-up: We thank the reviewer for the additional comments about the sample size, and we agree that it is always better to have n > 3 for statistical analysis. As described above, because of the complexity and the related limitation of animals, to first check if a radiotracer is worth further characterization its radiopharmaceutical properties for imaging alpha-synuclein in CNS, for each radiotracer, we performed 0-120 min dynamic scans for 2 or 3 to make sure the scan data are accountable. However, for the most promising radiotracer, we will further perform reproducibility in one animal for multiple times and in multiple animals. We understand the review’s concerns; however, we believe our data are sufficient to support our conclusions. Further characterizations such as blood input function, kinetic modeling, radiometabolite analysis, and larger sample size are necessary for our most promising leading radiotracer. We appreciate the reviewer’s constructive feedback and consider these suggestions critical for future optimization efforts.

Q12R: Please explore and furnish the results.

Q12 follow-up: We thank the reviewer for raising this important point regarding the potential influence of efflux transporters on [¹¹C]8i brain uptake. We agree that understanding whether [¹¹C]8i is a substrate of transporters such as P-glycoprotein (P-gp) or BCRP is critical for fully characterizing its CNS pharmacokinetics. However, as a tracer development study, our current stage focuses on understanding the fundamental molecular properties of our radioligands, we are focusing on characterization and identification of the most promising radioligands, and our reported studies meet our goal and support our conclusions that several of our radioligands are potent and specific towards α-synuclein with high brain uptakes. Our next step is to evaluate our leading compounds in disease models and identify the most promising compounds, once we determine the leading compounds, we will perform more detailed biochemical analysis to understand the impact of efflux transports and pharmacological properties.  Nonetheless, we appreciate the reviewer’s suggestion and will incorporate transporter interaction assays in subsequent studies to better define the compound’s pharmacological profile.

Q13R: Please study the stability and report any potential metabolites if any.

Q13 follow-up: We thank the reviewer for highlighting the importance of assessing radiotracer stability and metabolite profiles. We fully agree that such data are critical for evaluating the in vivo behavior and potential translational utility of PET tracers. However, due to the exploratory nature of the current study and time limitation in sample availability, we have not yet performed detailed radiometabolite analyses for [¹¹C]7f, [¹¹C]8i, or [¹¹C]7j during PET scans for imaging data acquisition. Nevertheless, the observed brain uptake patterns and washout kinetics from dynamic PET imaging are consistent with stable tracer behavior over the early scanning window. For the most promising radiotracer, our follow-up study will perform its radiometabolite analysis post-injection in the animal. We appreciate the reviewer’s constructive suggestion and view this as a key next step in our tracer development process.

Q14R: Revised version does not represent the times of the images in Fig 7d.

Q14 follow-up: As we mentioned in the method and figure legend, Figure 7d is the representative DVR images of tracer uptake using Logan reference based methods.

Q15R: Please include more animal per tracer and provide stats to be able to predict ideal responses.

Q15 follow-up: We have already addressed this in the Q11R. We thank the reviewer for the additional comments about the sample size, and we agree that it is always better to have n more than 3 for statistical analysis. However, as we mentioned, with limited number for non-human primate studies and safety concerns, it is very challenging to have each radiotracer PET more than 3 times at the current stage, which we are more focused on checking if a radiotracer is suitable for addition characterization of its radiopharmaceutical properties for α-synuclein in the brain. Please see our response above. We appreciate the reviewer’s constructive feedback and consider these suggestions critical for future optimization efforts.

Based on the responses provide by the authors, this research work needs additional experiments to determine the suitability of BBB penetration for the investigative compounds. Please perform the required experimentation and resubmit.

We appreciate the reviewer’s feedback. While we agree that additional animal experiments and kinetic modeling would strengthen current work. We had performed and provided the PET dynamic data that showed the brain uptake curve of each radiotracer, it also demonstrated if a radiotracer could enter the brain or not. We are not sure the question that we still need additional experimental to determine if the radiotracer is able to penetrate the BBB.  We respectfully believe the current data sufficiently address this question.

Round 3

Reviewer 2 Report

Comments and Suggestions for Authors

Q4R: It would have been great if the authors could have included control tissue to see any non α-synuclein or non Aβ fibrils as well.

Q4 follow up: We thank the reviewer for the thoughtful follow-up comment. We agree that including additional control tissues without α-synuclein and Aβ fibrils could provide complementary insight into the ligand binding selectivity. Our data indeed included young healthy control subjects in Figure 5 and 6 for more detailed characterization of our leading compounds. For Figure 4, the experiments focus on determining if α-synuclein potent compounds bind toward Aβ through a screening approach, we don’t think this impacts our decision if a ligand is selective binding toward alpha-synuclein over Aβ fibrilsFigures 5 and 6, clearly demonstrate that compounds have high selectivity for α-synuclein, evidenced by strong binding in PD tissues and minimal to no binding in both AD and young healthy control subjects.

Q10R: It would be ideal to include logP and PK study results as part of this manuscript with n=2 (2 NHPs per tracer) with stats.

Q10 follow-up: We agree with the reviewer including experimental LogP and pharmacokinetics (PK) be ideal. However, for the lead radiotracers, we had completed and presented the two hrs dynamic PET scan, the brain time tissue activity curves already showed whether the radiotracer was able to enter the brain and its brain washout pharmacokinetics. We understand the measurement of log P and PK is also able to predict a radiotracer can enter into the brain and its brain pharmacokinetics. In our future radiosynthesis of radiotracer for the brain, we will include this measurement as evidence of whether a radiotracer can enter the brain. Although we understand the more repeated PET scans we perform, the better we can confirm a radiotracer’s in vivo reproducibility. However, nonhuman primate brain 0-120 min dynamic scan is more complex than multiple scans for rodent brains. After a 0-120 dynamic scan, the monkey was required to take at least 4 weeks for recovery based on our institutional animal use and protection committee request. It is very challenging to have n > 4 for initially check whether a radiotracer can enter the brain and has suitable brain washout pharmacokinetics. However, for most promising radiotracer (s), we will perform multiple scans (n>4) to check their reproducibility for same animal and multiple animals prior to determining further toxicity and dosimetry validations.

Q11R: This needs to be addressed with required stats and n=2 or 3 per tracer.

Q11 follow-up: We thank the reviewer for the additional comments about the sample size, and we agree that it is always better to have n > 3 for statistical analysis. As described above, because of the complexity and the related limitation of animals, to first check if a radiotracer is worth further characterization its radiopharmaceutical properties for imaging alpha-synuclein in CNS, for each radiotracer, we performed 0-120 min dynamic scans for 2 or 3 to make sure the scan data are accountable. However, for the most promising radiotracer, we will further perform reproducibility in one animal for multiple times and in multiple animals. We understand the review’s concerns; however, we believe our data are sufficient to support our conclusions. Further characterizations such as blood input function, kinetic modeling, radiometabolite analysis, and larger sample size are necessary for our most promising leading radiotracer. We appreciate the reviewer’s constructive feedback and consider these suggestions critical for future optimization efforts.

Q12R: Please explore and furnish the results.

Q12 follow-up: We thank the reviewer for raising this important point regarding the potential influence of efflux transporters on [¹¹C]8i brain uptake. We agree that understanding whether [¹¹C]8i is a substrate of transporters such as P-glycoprotein (P-gp) or BCRP is critical for fully characterizing its CNS pharmacokinetics. However, as a tracer development study, our current stage focuses on understanding the fundamental molecular properties of our radioligands, we are focusing on characterization and identification of the most promising radioligands, and our reported studies meet our goal and support our conclusions that several of our radioligands are potent and specific towards α-synuclein with high brain uptakes. Our next step is to evaluate our leading compounds in disease models and identify the most promising compounds, once we determine the leading compounds, we will perform more detailed biochemical analysis to understand the impact of efflux transports and pharmacological properties. Nonetheless, we appreciate the reviewer’s suggestion and will incorporate transporter interaction assays in subsequent studies to better define the compound’s pharmacological profile.

Q13R: Please study the stability and report any potential metabolites if any.

Q13 follow-up: We thank the reviewer for highlighting the importance of assessing radiotracer stability and metabolite profiles. We fully agree that such data are critical for evaluating the in vivo behavior and potential translational utility of PET tracers. However, due to the exploratory nature of the current study and time limitation in sample availability, we have not yet performed detailed radiometabolite analyses for [¹¹C]7f, [¹¹C]8i, or [¹¹C]7j during PET scans for imaging data acquisition. Nevertheless, the observed brain uptake patterns and washout kinetics from dynamic PET imaging are consistent with stable tracer behavior over the early scanning window. For the most promising radiotracer, our follow-up study will perform its radiometabolite analysis post-injection in the animal. We appreciate the reviewer’s constructive suggestion and view this as a key next step in our tracer development process.

Q14R: Revised version does not represent the times of the images in Fig 7d.

Q14 follow-up: As we mentioned in the method and figure legend, Figure 7d is the representative DVR images of tracer uptake using Logan reference based methods.

Q15R: Please include more animal per tracer and provide stats to be able to predict ideal responses.

Q15 follow-up: We have already addressed this in the Q11R. We thank the reviewer for the additional comments about the sample size, and we agree that it is always better to have n more than 3 for statistical analysis. However, as we mentioned, with limited number for non-human primate studies and safety concerns, it is very challenging to have each radiotracer PET more than 3 times at the current stage, which we are more focused on checking if a radiotracer is suitable for addition characterization of its radiopharmaceutical properties for α-synuclein in the brain. Please see our response above. We appreciate the reviewer’s constructive feedback and consider these suggestions critical for future optimization efforts.

Based on the responses provide by the authors, this research work needs additional experiments to determine the suitability of BBB penetration for the investigative compounds. Please perform the required experimentation and resubmit.

We appreciate the reviewer’s feedback. While we agree that additional animal experiments and kinetic modeling would strengthen current work. We had performed and provided the PET dynamic data that showed the brain uptake curve of each radiotracer, it also demonstrated if a radiotracer could enter the brain or not. We are not sure the question that we still need additional experimental to determine if the radiotracer is able to penetrate the BBB. We respectfully believe the current data sufficiently address this question.

Please revise fig 6b (please make 3 independent experiments and show the error bars) and 7d (with time range, for example, what time range the image is correlated to) and update the manuscript as minor revisions.

Author Response

Reviewer 2:

Please revise Figure 6b (please perform 3 independent experiments and show the error bars) and Figure 7d (indicate the time range, e.g., what time range the image corresponds to) and update the manuscript accordingly as minor revisions

We sincerely thank the reviewer for these thoughtful and constructive suggestions, which reflect a high level of expertise in this area.

Regarding Figure 6b:

We thank the reviewer for this helpful comment. The assays shown in Figure 6b were performed by our collaborators under the supervision of Dr. Chester A. Mathis, who retired last year. According to their records, the experiments were conducted in triplicate, and the plotted values are the mean of three independent measurements. Unfortunately, retrieving the original replicate data has been difficult due to the circumstances, but we are actively working to obtain them and will provide error bars if possible. In the meantime, we have added a note to the figure legend clarifying that the data represent mean values from three independent experiments. We appreciate the reviewer’s understanding.

Regarding Figure 7d:

We thank the reviewer for this helpful suggestion. In response, we have clarified in the Methods section that voxel-wise reference Logan modelling was performed with a linear start time of 30–120 minutes using the cerebellum as the reference region. We have also added the corresponding equation to illustrate the calculation. In addition, the Figure 7 caption has been updated to specify the modelling approach and the time range.

We hope these clarifications address the reviewer’s comments. We greatly appreciate the reviewer’s expertise and valuable feedback, which have helped us improve the clarity and rigor of the manuscript.